

# Mechanism of ozone loss under enhanced water vapour conditions in the mid-latitude lower stratosphere in summer

Sabine Robrecht[1], Bärbel Vogel[1], Jens-Uwe Grooß[1], Karen Rosenlof[2], Troy Thornberry[2,3],
Andrew Rollins[2], Martina Krämer[1], Lance Christensen[4], and Rolf Müller[1]

[1]Forschungszentrum Jülich, Institute of Energy and Climate Research (IEK-7), Jülich, Germany
[2]NOAA Earth System Research Laboratory (ESRL) Chemical Sciences Division, Boulder, CO 80305 USA
[3]University of Colorado, Cooperative Institute for Research in Environmental Sciences, Boulder, CO 80309 USA
[4]California Institute of Technology, Jet Propulsion Laboratory, Pasadena, CA 91125 USA

*Correspondence to:* Sabine Robrecht (sa.robrecht@fz-juelich.de)

**Abstract.** Water vapour convectively injected into the mid-latitude lowermost stratosphere could affect stratospheric ozone. The associated potential ozone loss process requires low temperatures and an elevated water vapour mixing ratio. An increase in sulphate aerosol surface area due to a volcanic eruption or geoengineering could increase the likelihood of occurrence of this process. However, the chemical mechanism of this ozone loss process has not yet been analysed in sufficient detail and its

sensitivity to various conditions is not yet clear. Under conditions of climate change associated with an increase in greenhouse gases, both a stratospheric cooling and an increase in water vapour convectively injected into the stratosphere is expected. Understanding the influence of low temperatures, elevated water vapour and enhanced sulphate particles on this ozone loss mechanism is a key step in estimating the impact of climate change and potential sulphate geoengineering on mid-latitude ozone.

Here, we analyse the ozone loss mechanism and its sensitivity to various stratospheric conditions in detail. Conducting a box-model study with the Chemical Lagrangian Model of the Stratosphere (CLaMS), chemistry was simulated along a 7-day backward trajectory. This trajectory was calculated neglecting mixing of neighbouring air masses. Chemical simulations were initialized using measurements taken during the Study of Emissions and atmospheric Composition, Clouds and Climate Coupling by Regional Surveys (SEAC[4]RS) aircraft campaign (2013, Texas), which encountered an elevated water vapour

mixing ratio at a pressure level around 100 hPa. We present a detailed analysis of the ozone loss mechanism, including the chlorine activation, chlorine catalysed ozone loss cycles, maintenance of activated chlorine and the role of active nitrogen oxide radicals ($NO_x$). Focussing on a realistic trajectory in a temperature range from 197–203 K, a threshold in water vapour of 11.0–11.6 ppmv has to be exceeded and maintained for stratospheric ozone loss to occur. We investigated the sensitivity of the water vapour threshold to temperature, sulphate content, inorganic chlorine ($Cl_y$), inorganic nitrogen ($NO_y$) and inorganic

bromine ($Br_y$). The water vapour threshold is mainly determined by the temperature and sulphate content. However, the amount of ozone loss depends on $Cl_y$, $NO_y$, $Br_y$ and the duration of the time period over which chlorine activation can be maintained. Our results show that to deplete ozone, a chlorine activation time of 24 to 36 hours for conditions of the water vapour threshold with low temperatures and high water vapour mixing ratios must be maintained. A maximum ozone loss of 9% was found for a 20 ppmv water vapour mixing ratio at North American Monsoon (NAM) tropopause standard conditions with the model run





along a realistic trajectory. For the same trajectory, using observed conditions (of 10.6 ppmv), whether ozone loss occurs was simulated dependent on the sulphate amount assumed. Detailed analysis of current and future possibilities is needed to assess whether enhanced water vapour conditions in the summertime mid-latitude lower stratosphere leads to significant ozone loss.

## 1 Introduction

The impact of water vapour convectively injected into the lowermost stratosphere on the mid-latitude ozone layer is a matter of current debate (Anderson et al., 2012, 2017; Ravishankara, 2012; Schwartz et al., 2013). Borrmann et al. (1996, 1997) and Solomon et al. (1997) investigated the influence of cirrus clouds on ozone chemistry in the lowermost stratosphere. Anderson et al. (2012) proposed a potential ozone depletion in the mid-latitude stratosphere in summer under conditions of enhanced water vapour and low temperatures. They proposed chemical ozone loss to occur under these conditions through processes related to ozone loss known from polar regions early spring (e.g. Grooß et al., 2011; Solomon, 1999; Vogel et al., 2011). Here, we present a detailed analysis of this ozone loss mechanism and an extensive investigation of its sensitivity to a variety of conditions.

In the bulk and on the surface of cold condensed stratospheric particles, such as binary $H_2SO_4/H_2O$ solutions, ternary solutions, NAT (Nitric Acid Trihydrate) and ice particles (e.g. Spang et al., 2018), inactive chlorine species (HCl, $ClONO_2$) can be converted to active chlorine ($ClO_x$=Cl+ClO+2 × $Cl_2O_2$) through the heterogeneous reactions R1, R2 and R3 (Solomon et al., 1986; Prather, 1992; Crutzen et al., 1992) and the subsequent photolysis of $Cl_2$ and HOCl.

$$ClONO_2 + HCl \rightarrow HNO_3 + Cl_2 \tag{R1}$$

$$ClONO_2 + H_2O \rightarrow HNO_3 + HOCl \tag{R2}$$

$$HCl + HOCl \rightarrow H_2O + Cl_2 \tag{R3}$$

The heterogeneous reactions R1 and R2 drive the conversion of active nitrogen-oxides ($NO_x$) into $HNO_3$. After chlorine activation, catalytic ozone loss cycles can occur, such as the ClO-dimer-cycle (Molina and Molina, 1987, **C1**) and the ClO-BrO-cycle (McElroy et al., 1986, **C2**). These cycles are responsible for the rapid ozone loss observed in Antarctic spring (e.g. Solomon, 1999).

ClO-Dimer-Cycle (**C1**):

$$ClO + ClO + M \rightarrow ClOOCl + M \tag{R4}$$

$$ClOOCl + h\nu \rightarrow Cl + ClOO \tag{R5}$$

$$ClOO + M \rightarrow Cl + O_2 + M \tag{R6}$$

$$2x(Cl + O_3 \rightarrow ClO + O_2) \tag{R7}$$

$$\text{net: } 2O_3 + h\nu \rightarrow 3O_2 \tag{C1}$$



ClO-BrO-Cycle (**C2**):

$$ClO + BrO \rightarrow Br + Cl + O_2 \tag{R8}$$

$$Br + O_3 \rightarrow BrO + O_2 \tag{R9}$$

$$Cl + O_3 \rightarrow ClO + O_2 \tag{R7}$$

$$\text{net: } 2\,O_3 \rightarrow 3\,O_2 \tag{C2}$$

In a third catalytic ozone loss cycle, **C3**, HOCl is formed and subsequently photolysed yielding OH and Cl radicals leading to stratospheric ozone destruction. This cycle was originally proposed by Solomon et al. (1986) as an ozone depleting cycle in the Antarctic lower stratosphere, but for polar ozone destruction, this cycle turned out to be of minor importance (Solomon, 1999). Nevertheless, **C3** would be expected to play a role in ozone loss in the mid-latitude lower stratosphere (e.g. Daniel et al., 1999;

Ward and Rowley, 2016).

$$ClO + HO_2 \rightarrow HOCl + O_2 \tag{R10}$$

$$HOCl + h\nu \rightarrow Cl + OH \tag{R11}$$

$$OH + O_3 \rightarrow HO_2 + O_2 \tag{R12}$$

$$Cl + O_3 \rightarrow ClO + O_2 \tag{R7}$$

$\text{net: } 2\,O_3 + h\nu \rightarrow 3\,O_2$        (**C3**)

Under the very dry conditions in the polar stratosphere, very low temperatures (below ∼195 K) are required for heterogeneous chlorine activation through reactions R1–R3 (Solomon, 1999; Shi et al., 2001). An enhancement of water vapour above background values would allow chlorine activation at higher temperatures (200–205 K) (Drdla and Müller, 2012), which led to the hypothesis hat chlorine activation and subsequent ozone loss could occur at mid-latitudes in summer in the lowermost

stratosphere (Anderson et al., 2012, 2017; Anderson and Clapp, 2018).

An enhanced stratospheric sulphate aerosol content increases heterogeneous chlorine activation by increasing the surface area of the condensed particles (Drdla and Müller, 2012; Solomon, 1999). As an example, the aerosol surface area density in the lower stratosphere ranges between ∼0.5 and 1.5 $\mu$m$^2$cm$^{-3}$ under non-volcanic conditions (Thomason and Peter, 2006), while the perturbation of Mt. Pinatubo yielded peak values of more than 40 $\mu$m$^2$cm$^{-3}$ (Thomason et al., 1997). In the stratosphere,

water vapour increases with altitude, primarily due to methane oxidation (LeTexier et al., 1988; Rohs et al., 2006). The upper branch of the Brewer Dobson circulation (BDC) transports higher stratospheric water vapour mixing ratios down to lower altitudes at mid to high latitudes, and this air mixes with the low water vapour containing air from the tropics that has moved poleward in the lower branch of the BDC (e.g. Brewer, 1949; Randel et al., 2004; Schwartz et al., 2013; Konopka et al., 2015; Poshyvailo et al., 2018), giving typical mid-latitude lowermost stratosphere values of 2–6 ppmv H$_2$O.

However, above North America in summer enhanced water vapour mixing ratios of 10–18 ppmv at an altitude of ∼16.5 km (380 K potential temperature, ∼ 100 hPa) (Smith et al., 2017) have been observed, which were connected with deep convective



storm systems penetrating the tropopause (Homeyer et al., 2014; Herman et al., 2017; Smith et al., 2017). These convective overshooting events can transport ice crystals into the lowermost stratosphere, where the ice evaporates leading to a local enhancement of water vapour (Hanisco et al., 2007; Schiller et al., 2009; Herman et al., 2017).

As greenhouse gases increase, models predict that more water may be convectively transported into the stratosphere (Trapp
et al., 2009; Klooster and Roebber, 2009). This increases the possibility that the ozone loss process proposed by Anderson et al. (2012) will occur, especially in the case of an additional enhancement of stratospheric sulphate particles caused by volcanic eruptions or sulphate geoengineering. However, the occurrence of this ozone loss process requires halogens to be present, which are decreasing in the stratosphere due to the Montreal Protocol ans its amendments and adjustments (WMO, 2014). For estimating the impact of both climate change and a possible sulphate geoengineering on the mid-latitude ozone layer, it
is necessary to consider the influence of enhanced water vapour and sulphate content on mid-latitude ozone chemistry in the lowermost stratosphere in more detail.

In the study by Anderson et al. (2012), a range of initial mixing ratios for HCl and $ClONO_2$ with rather high concentrations of 850 pptv HCl and 150 pptv $ClONO_2$ was assumed. Here, we investigate ozone loss in mid-latitude summer based on measurements from flights by the NASA ER-2 aircraft during the Studies of Emissions and Atmospheric Composition, Clouds and
Climate Coupling by Regional Surveys (SEAC[4]RS) campaign, which was based in Houston, Texas in 2013 (Toon et al., 2016). Conducting box-model simulations with the Chemical Lagrangian Model of the Stratosphere (CLaMS, McKenna et al., 2002a, b), the ozone loss mechanism is analysed in greater detail. The model setup is described in Section 2. In Sec. 3, the chlorine activation step, catalytic ozone loss cycles and the maintenance of activated chlorine levels in the mid-latitude stratosphere are investigated in detail. The sensitivity of this mechanism to water vapour, sulphate content, temperature, $Cl_y$ mixing ratio
($Cl_y$=HCl+$ClONO_2$+$ClO_x$), reactive nitrogen ($NO_y$=$NO_x$+$HNO_3$ $2N_2O_5$+$NO_3$) and inorganic bromine ($Br_y$) is explored in Sec. 4. Case studies, which extend the simulated time period and assume conditions based on both SEAC[4]RS and MACPEX (Mid-latitude Airborne Cirrus Properties Experiment) measurements as well as conditions used in the study of Anderson et al. (2012) further illustrate these sensitivities in Sec. 5.

## 2  Model setup

The simulations presented here were performed with the box-model version of CLaMS (McKenna et al., 2002a, b). Stratospheric chemistry is simulated based on a setup used in previous studies (Grooß et al., 2011; Müller et al., 2018; Zafar et al., 2018) for single air parcels along trajectories including diabatic descent and neglecting mixing between neighbouring air masses. A full chemical reaction scheme comprising gas phase and heterogeneous chemistry is applied using the SVODE-solver (Brown et al., 1989). Chemical reaction kinetics are taken from Sander et al. (2011), heterogeneous reaction rates for
R1–R3 were calculated based on the the study of Shi et al. (2001), and photolysis rates are calculated for spherical geometry (Becker et al., 2000). In contrast to the setup in Grooß et al. (2011), Müller et al. (2018) and Zafar et al. (2018), only formation of liquid particles (both binary $H_2O$/$H_2SO_4$ and ternary $HNO_3$/$H_2O$/$H_2SO_4$ solutions) is allowed (i.e. no NAT or ice particles



are formed in this model setup). Note that this is also different from the study of and to the study of Borrmann et al. (1996, 1997), who investigated lowermost stratospheric ozone chemistry on cirrus clouds.

## 2.1 Measurements

The box model simulations were initialized using water vapour, ozone and $CH_4$ measurements taken during the SEAC$^4$RS aircraft campaign and water vapour, ozone and $N_2O$ measurements taken during MACPEX campaign (more information on the chemical initialization is provided in Section 2.3). The investigation of mid-latitude ozone chemistry presented here is primarily based on SEAC$^4$RS conditions, while simulations based on MACPEX data were conducted to complement the results of the investigation by showing a further measurement-based example. The SEAC$^4$RS campaign took place during the North American summertime, while the MACPEX campaign was in spring, prior to the build up of the North American Monsoon (NAM) anticyclone.

The SEAC$^4$RS campaign was based in Houston, Texas, and took place during August and September 2013 (Toon et al., 2016). One aim of this campaign was to investigate the impact of deep convective clouds on the water vapour content and the chemistry in the lowermost stratosphere. We initialized the model using measurements taken on 8 August 2013 by the Harvard Lyman-a photofragment fluorescence hygrometer (HWV, Weinstock et al., 2009), which flew on the NASA ER-2 high altitude research aircraft. Ozone was initialized in our simulations, using $O_3$ measurements from the National Oceanic and Atmospheric Administration (NOAA) UAS-$O_3$-instrument (Gao et al., 2012). Initial $Cl_y$ and $NO_y$ were determined using tracer-tracer correlations (for more informations see Sec. 2.3) based on methane measurements with the Harvard University Picarro Cavity Ring down Spectrometer (HUPCRS) (Werner et al., 2017).

The simulation results initialized with SEAC$^4$RS measurements were compared with a case of enhanced lower stratospheric water sampled during the spring 2011 MACPEX campaign (Rollins et al., 2014) also based in Houston, Texas. The water vapour values used here were measured by the Fast In-situ Stratospheric Hygrometer (FISH), which employs the Lyman-$\alpha$ photofragment fluorescence technique (Meyer et al., 2015). MACPEX ozone was measured by the UAS-$O_3$ instrument (Gao et al., 2012). Initial $Cl_y$ and $NO_y$ were assumed based on tracer-tracer correlations with $N_2O$ that was measured by the Jet Propulsion Laboratory's Aircraft Laser Infrared Absorption Spectrometer (ALIAS) instrument (Webster et al., 1994).

## 2.2 Trajectories

Diabatic trajectories were calculated using wind and temperature data from the ERA-Interim reanalysis (Dee et al., 2011) with $1° \times 1°$ resolution provided by the European Centre for Medium-Range Weather Forecasts (ECMWF). The vertical velocities were calculated from the total diabatic heating rates derived from ERA-Interim data (Ploeger et al., 2010). Trajectories (7 day forward and backward) were initialized at locations during MACPEX and SEAC$^4$RS where stratospheric water vapour was over 10 ppmv.

Selected examples of calculated trajectories are shown in Fig. 1. These trajectories were chosen for the chemical analysis, because their initial conditions exhibited enhanced water vapour relative to the overall background and the temperatures were very low. These are then the most suitable for the occurrence of the mechanism proposed by Anderson et al. (2012). In the left





panel, backward trajectories are presented in the range of $-7$ to $0$ days from the time of measurement and forward trajectories in the range from $0$ to $7$ days. In the right panel, the location of the measurement is shown by a red square. The black trajectory refers to a measurement on 11 April 2011 during the MACPEX campaign. The potential temperature level of this trajectory is around 380 K and above the tropopause located at $\sim$350 K, which was deduced from the temperature profile measured during

the flight on 11 April 2011. The backward trajectory reaches very low temperatures with a minimum temperature of 191 K. The forward trajectory shows a strongly increasing temperature and pressure level due to a decrease in altitude. Coming from the Western Pacific, this air parcel passes the North American continent briefly. For this study numerous water measurements of the MACPEX campaign were analysed, but only few values of more than 10 ppmv were observed above the tropopause. The trajectory marked in blue in Fig. 1 is based on measurements on 8 August 2013 during the SEAC$^4$RS campaign. With a

potential temperature of 380 to 390 K, this trajectory is above the tropopause of $\sim$370 K, deduced from the temperature profile measured during the flight. Both, the forward and backward trajectories stay in the region of the North American continent. For the SEAC$^4$RS campaign, the temperature range of the backward trajectory varies between 197 and 202 K and the forward trajectory exhibits increasing temperatures. In addition, we considered trajectories based on other SEAC$^4$RS measurements with enhanced water vapour, however most of them exhibit higher mean temperatures of at least 200 K. Since low temperatures

are expected to push stratospheric ozone depletion in mid-latitudes (Anderson et al., 2012) due to faster heterogeneous chemical reactions and thus faster chlorine activation, the SEAC$^4$RS backward trajectory (Fig.1, blue, day $-7$ to $0$) is selected here as the standard trajectory. This trajectory is used to analyse the chemical mechanisms affecting lower stratospheric ozone under various water vapour conditions, and to explore the sensitivity of these processes to different initial conditions.

## 2.3   Initialization

Important trace gases for ozone chemistry – $O_3$, $Cl_y$ and $NO_y$ – are initialized based on measurements during the SEAC$^4$RS and MACPEX aircraft campaigns over North America (see Sec. 2.1). Ozone and water vapour were measured directly during the aircraft campaigns, $Cl_y$ and $NO_y$ are inferred from tracer-tracer relations using either $CH_4$ (SEAC$^4$RS) or $N_2O$ (MACPEX) measured on the aircraft employed. The initialization of all further trace gases except of water vapour were taken from the full chemistry 3D-CLaMS simulation (Vogel et al., 2015, 2016) for summer 2012 at the location of the measurement. Chemistry

was initialized 7 days before the measurement. However, this time shift does not affect the sensitivities and the mechanism investigated here, because the trace gases $Cl_y$ and $NO_y$ were initialized based on measured $CH_4$ and $N_2O$ mixing ratios, which are not significantly changing during a 7-day box-model simulation.

### 2.3.1   Standard case

In the standard case, the initial values of $O_3$, $Cl_y$ and $NO_y$ are determined based on an observation with an enhanced water

vapour content of 10.6 ppmv (measured by the HWV-instrument) from the SEAC$^4$RS (Toon et al., 2016) aircraft campaign. A gas phase equivalent mixing ratio for background sulphuric acid ($H_2SO_4$) of 0.20 ppbv is assumed. Initial CO (49.6 ppbv) is taken from the 3D-CLaMS simulation (Vogel et al., 2015), which is higher than the measured value of 34.74 ppbv (measured by the HUPCRS instrument). Simulations assuming the measured CO mixing ratio showed only a minor difference to the



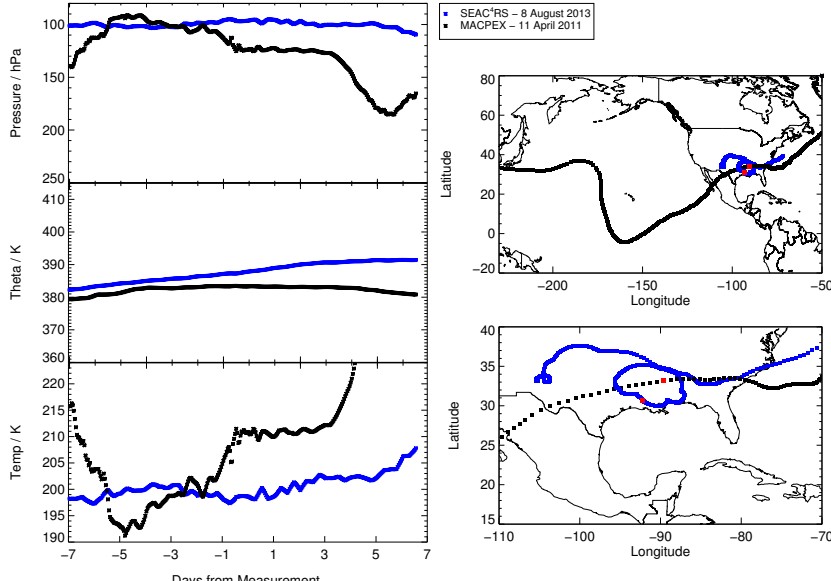

**Figure 1.** Pressure, potential temperature, temperature and location of selected trajectories calculated based on measurements with enhanced water vapour during the MACPEX (black) and SEAC⁴RS (blue) aircraft campaigns. Red squares mark the location of the measurement (right). For the right panels the bottom panel exhibits a zoom from the top panel. In the bottom panel, the MACPEX trajectory consists of single squares due to a faster movement of the air parcel in that region.

results presented here. The initial values for the main trace gases for the standard case are summarized in Table 1. Note, in the 3D-CLaMS simulation, the mixing ratios of HCl (131 ppt, CLaMS), $O_3$ (206 ppb, CLaMS) and $HNO_3$ (354 ppt, CLaMS) are at the location of the SEAC⁴RS measurement lower than in the standard initialization (see Tab. 1).

Since $Cl_y$ and $NO_y$ were not measured during the SEAC⁴RS ER2-flights in the lowermost stratosphere, values for $Cl_y$ and

5 $NO_y$ are calculated through tracer-tracer correlations (Grooß et al., 2014, see Appendix A for equations) based on a SEAC⁴RS $CH_4$ measurement (of 1.776 ppmv) on 8 August 2013. The $Cl_y$-$CH_4$ correlation was calculated from measurements of the Airborne Chromatograph for Atmospheric Trace Species (ACATS) during flights of the ER-2 aircraft and from measurements by the cryogenic whole air sampler (Triple) during balloon flights at mid and high latitudes in the year 2000 (Grooß et al., 2002). Between the year 2000 and 2013 stratospheric $CH_4$ increased and $Cl_y$ decreased. Hence, the change of both lowermost

stratospheric $CH_4$ and $Cl_y$ has to be taken into account when using this tracer-tracer correlation. The increase in $CH_4$ was estimated to be equivalent to the growth rate for tropospheric $CH_4$. This growth rate was calculated to be 45.8 ppb from the year 2000 to 2013 by determining and adding every annual mean of the tropospheric $CH_4$ growth rate given in GHG Bulletin (2014). Subtracting this increase of $CH_4$ from the measured $CH_4$ mixing ratio yields an equivalent-$CH_4$ for the year 2000. From the $CH_4$ equivalent, an equivalent $Cl_y$ mixing ratio for the year 2000 was calculated using the tracer-tracer correlation

(Grooß et al., 2014). The annual decrease of $Cl_y$ is assumed to be 0.8% (WMO, 2014) from the year 2000 to 2013, and thus



the initial $Cl_y$ is calculated to be 156 ppt.

Initial $NO_y$ was calculated through a $N_2O$ correlation. Since no $N_2O$ was measured on the ER2-flights during SEAC$^4$RS, stratospheric N2O was first estimated through a methane correlation (Grooß et al., 2002), which is based on measurements from the year 2000. Hence, the equivalent $CH_4$ mixing ratio for the year 2000 (see above) was used to calculate an $N_2O$ equivalent. Considering an estimated increase in $N_2O$ of 10.4 ppb from 2000 to 2013, which was determined in the same way as the $CH_4$ change (GHG Bulletin, 2014), the $N_2O$ mixing ratio related to the time of the measurement in 2013 was calculated. Afterwards $NO_y$ is calculated with a correlation from Grooß et al. (2014) to be 782.9 ppt.

This standard case initialization is shown in Table 1. Because of the uncertain conditions in convective overshooting plumes, sensitivity box-model simulations are conducted. Furthermore, testing the impact of various parameters on chemical ozone loss is intended to yield a better understanding of the balance between stratospheric ozone production and ozone loss, which is a key aspect for potential mid-latitude ozone depletion. The assumed water vapour content in a simulation is varied from 5 to 20 ppmv. In addition, simulations assuming the same water vapour range and a constant temperature in a range from 195–220 K are conducted assuming sulphate background conditions with a gas phase equivalent of 0.20 ppbv and 10×enhanced sulphate (2.00 ppbv) for illustrating the dependence of ozone loss on water vapour and temperature. Furthermore, sensitivity simulations are conducted, assuming 80% $Cl_y$, 80% $NO_y$ or 50% $Br_y$, and a standard case simulation along a 19-day trajectory is calculated.

**Table 1.** Mixing ratios and sources used for initialization of relevant trace gases. The standard initialization is based on SEAC$^4$RS measurements. $Cl_y$ and $NO_y$ values were determined based on tracer-tracer correlations (see text) for the standard and the MACPEX case. The high $Cl_y$ case is based on Fig. 2 from Anderson et al. (2012). Initial mixing ratios of $ClO_x$ species were assumed to be zero for all cases.

| Species | Standard case | | | Case of high $Cl_y$ | | MACPEX case | |
| --- | --- | --- | --- | --- | --- | --- | --- |
| | Value | Source | Sensitivity simulation | Value | Source | Value | Source |
| $O_3$ | 303.2 ppbv | UAS-$O_3$ | | 303.2 ppbv | UAS-$O_3$ | 283.0 ppbv | UAS-$O_3$ |
| $CH_4$ | 1.76 ppmv | CLaMS-3D | | 1.76 ppmv | CLaMS-3D | 1.68 ppmv | CLaMS-3D |
| CO | 49.6 ppbv | CLaMS-3D | | 49.6 ppbv | CLaMS-3D | 19.0 ppbv | CLaMS-3D |
| HCl | 149.5 pptv | tracer corr. | 80% $Cl_y$ | 850 pptv | Anderson et al. (2012) | 52.7 pptv | tracer corr. |
| $ClONO_2$ | 6.2 pptv | tracer corr. | | 150 pptv | Anderson et al. (2012) | 2.19 pptv | tracer corr. |
| $HNO_3$ | 439.23 pptv | tracer corr. | 80% $NO_y$ | 1.19 ppbv | see section 2.3.2 | 390.3 pptv | tracer corr. |
| NO | 144.8 pptv | tracer corr. | | 325 pptv | Anderson et al. (2012) | 114.6 pptv | tracer corr. |
| $NO_2$ | 144.8 pptv | tracer corr. | | 375 pptv | Anderson et al. (2012) | 114.6 pptv | tracer corr. |
| $Br_y$ | 6.9 pptv | CLaMS-3D | 50% $Br_y$ | 6.9 pptv | CLaMS-3D | 1.2 pptv | CLaMS-3D |
| $H_2O$ | 5–20 ppmv | | 5–20 ppmv | 5–20 ppmv | | 5–20 ppmv | |
| $H_2SO_4$ | 0.2 ppbv, 0.6 ppbv, 2.0 ppbv | | | 0.2 ppbv, 0.6 ppbv | | 0.2 ppbv, 0.6 ppbv | |
| Temperature | 195–220 K | | const. temp | | | | |





### 2.3.2 Case of high $Cl_y$

Simulations conducted assuming high $Cl_y$ and $NO_y$ concentrations taken from Fig. 2 in Anderson et al. (2012) are referred to as "Case of high $Cl_y$", which constitutes a worst case scenario. In the case of high $Cl_y$, $HNO_3$ is determined as 1.19 ppb assuming the same ratio for $HNO_3$ (63% of total $NO_y$) and $NO+NO_2$ (37% of total $NO_y$) as in the standard case. An overview

of the important trace gases in the initialization is given in Tab. 1. The results of the case initialize with high $Cl_y$ are compared with the results obtained from standard case simulations.

### 2.3.3 MACPEX Case

Similar to the standard case, a simulation based on measurements during the MACPEX aircraft campaign 2011 (Rollins et al., 2014) was conducted, referred to as "MACPEX Case". This case presents a further example for an event with high stratospheric

water vapour based on airborne measurements and complements the results obtained from the standard case. All trace gases except for ozone, $Cl_y$, $NO_y$ and water vapour are taken from a 3D-CLaMS-simulation (Vogel et al., 2015, 2016). Initial $Cl_y$ and $NO_y$ is calculated based on $N_2O$ and $CH_4$ tracer-tracer correlations (Grooß et al., 2014) with corrections considering a $N_2O$ increase from 2009 to 2013 and a $CH_4$ increase from 2000 to 2009. $Cl_y$ is determined using the same correlation with $CH_4$ as for the standard case. Therefore $CH_4$ is first calculated using measured $N_2O$ of 320.28 ppbv and a correlation based

on measurements from 2009 (Grooß et al., 2014). The increase of stratospheric $CH_4$ and $N_2O$ is considered as described for the standard case (GHG Bulletin, 2014). First, an increase in $N_2O$ of 1.6 ppb from 2009 to 2011 is estimated to adjust $N_2O$. Furthermore calculated $CH_4$ is adjusted considering a difference between $CH_4$ in 2000 and 2009 of 0.026 ppm. The annual decrease of $Cl_y$ from 2000 to 2011 is assumed to be 0.8% (WMO, 2014). A summary of the initial values for main tracers assumed in the MACPEX case are given in Table 1.

## 3    Mid-latitude ozone chemistry

Mid-latitude ozone chemistry in the lowermost stratosphere depends on water vapour abundance and temperature. This study focuses on the water vapour dependence of stratospheric ozone chemistry by analysing chemical processes occurring in a box-model simulation along a realistic trajectory in the temperature range from 197-203 K under several water vapour conditions.

In Figure 2 the mixing ratio of ozone, $ClO_x$ and $NO_x$ is shown for two simulations assuming 5 ppmv (dashed line) and 15 ppmv (solid line) $H_2O$. For the low water vapour (5 ppmv) case, net ozone formation occurs, the $ClO_x$ mixing ratio remains low and the $NO_x$ mixing ratio high. In contrast, assuming a water vapour mixing ratio of 15 ppmv, ozone depletion accompanied by a decrease in $NO_x$ occurs, coupled with chlorine activation as indicated by the increasing $ClO_x$ mixing ratio. The sensitivity to variations in water vapour conditions on stratospheric ozone is tested here by conducting simulations with standard conditions

but varying the assumed water vapour mixing ratio from 5 to 20 ppmv in varying increments, with the resolution increased near the changeover from ozone production to destruction.





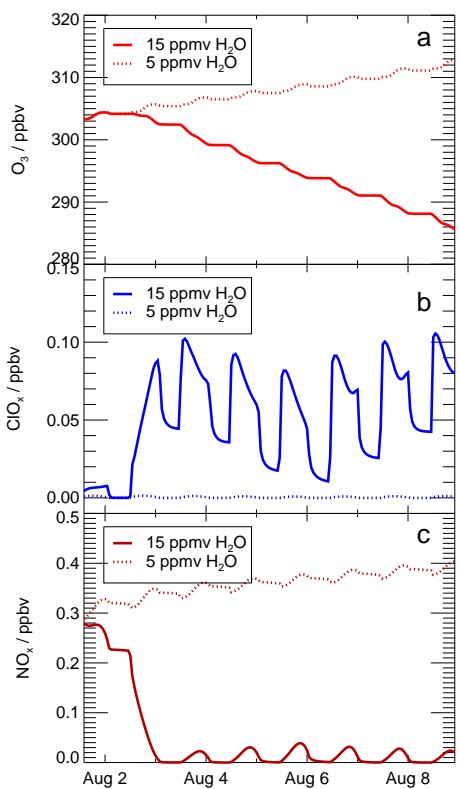

**Figure 2.** Volume mixing ratio of $O_3$ (panel a), $ClO_x$ (panel b) and $NO_x$ (panel c) during a simulation with 15 ppmv $H_2O$ and 5 ppmv $H_2O$.

In Figure 3, the ozone values reached at the end of the 7-day simulation (end ozone, blue squares) are plotted as a function of the assumed water vapour mixing ratio. The initial ozone value, of 303.2 ppbv, is shown by the grey line. Blue squares lying above that line are cases with ozone production, those lying below that line are cases with ozone destruction. The threshold is determined as the water vapour mixing ratio at which the end ozone value clearly falls below the end ozone that is reached for

5    low water vapour amounts. For the standard case shown in Fig. 3, this threshold is reached at a water vapour mixing ratio in the range of 11.0 to 11.6 ppmv. By 12 ppmv of water vapour, the system is clearly in an ozone destruction regime. The occurrence of the water vapour threshold and ozone depletion is related to chlorine activation. The time until chlorine activation occurs in the simulation is plotted in Fig. 3 as violet triangles. Assuming that chlorine activation occurs when the $ClO_x$ mixing ratio exceeds 10% of total $Cl_y$ (Drdla and Müller, 2012), plotted here is the time when chlorine activation first occurs in the model.

10    Since the $ClO_x/Cl_y$ ratio is dependent on the diurnal cycle, the 24-hours mean value of the $ClO_x$ mixing ratio was used to determine the chlorine activation time. For low water vapour mixing ratios, no chlorine activation time is plotted, because no chlorine activation occurs. Chlorine activation only occurs when the water vapour threshold is exceeded (Fig. 3). Near the water



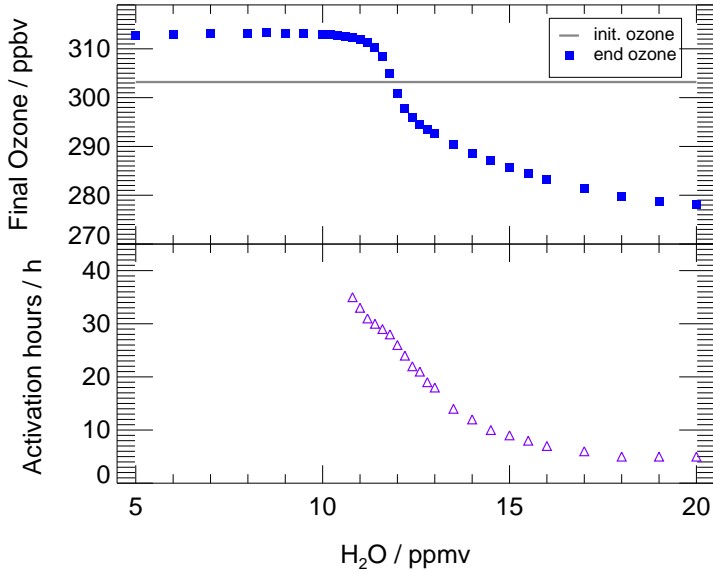

**Figure 3.** Impact of the water vapour content on the ozone mixing ratio (final ozone, blue squares) reached at the end of the 7-day simulation along the standard trajectory and assuming standard conditions. The initial ozone amount is marked by the grey line. In the bottom panel, violet triangles show the time until chlorine activation occurs is. For low water vapour mixing ratios no chlorine activation occurs.

vapour threshold, the activation time is 24 to 36 hours and it decreases with increasing water vapour mixing ratios. It requires 5 hours at 20 ppmv $H_2O$. The shorter the chlorine activation time, the longer activated chlorine exists during the simulation yielding greater ozone depletion. The processes yielding ozone depletion at high water vapour conditions as well as ozone formation at low water vapour are analysed in detail in the subsequent sections. For this investigation we use the simulated

5   reaction rates for each chemical reaction along the course of the calculation. For high water vapour mixing ratios the roles of both chlorine activation and a decrease in the $NO_x$ mixing ratio (Fig. 2) are discussed.

### 3.1   Ozone formation at low water vapour mixing ratios

At water vapour mixing ratios up to 11.8 ppmv, ozone formation occurs (see Fig. 3). This ozone formation is mainly driven by the photolysis of $O_2$

10   $$O_2 + h\nu \rightarrow 2\,O(^3P) \tag{R13}$$

and the subsequent reaction

$$O_2 + O(^3P) + M \rightarrow O_3 + M. \tag{R14}$$





Additionally, photolysis of $NO_2$

$$NO_2 + h\nu \rightarrow NO + O(^3P) \tag{R15}$$

followed by R14 leads to ozone formation. NO radicals, which are formed in R15, mainly react with ozone as well as ClO and BrO forming Cl and Br radicals.

$$NO + O_3 \rightarrow O_2 + NO_2 \tag{R16}$$
$$ClO + NO \rightarrow Cl + NO_2 \tag{R17}$$
$$BrO + NO \rightarrow Br + NO_2 \tag{R18}$$

Since these radicals react with ozone in reaction R7 ($Cl + O_3$) and R9 ($Br + O_3$), not all of the $O(^3P)$ formed in R15 yields a net ozone formation. However, R15 is part of the "Ozone Smog Cycle" (Haagen-Smit, 1952) known from tropospheric chemistry,
which has also an impact on stratospheric chemistry (Grenfell et al., 2006; Grooß et al., 2011).

$$OH + CO \rightarrow CO_2 + H \tag{R19}$$
$$H + O_2 + M \rightarrow HO_2 + M \tag{R20}$$
$$HO_2 + NO \rightarrow NO_2 + OH \tag{R21}$$
$$NO_2 + h\nu \rightarrow NO + O(^3P) \tag{R15}$$
$$O_2 + O(^3P) + M \rightarrow O_3 + M \tag{R14}$$
$$\text{net: } CO + 2O_2 \rightarrow CO_2 + O_3 \tag{C4}$$

The rate of this cycle is determined by reaction R19 at low water vapour mixing ratios, and its net reaction is the oxidation of CO. The ozone formation through this cycle contributes around 40% to the total ozone formation at 5 ppmv in our box model standard simulation. Hence, the ozone formation which occurs in the simulations assuming low water vapour mixing ratios is
due to both the photolysis of $O_2$ and cycle **C4**.

### 3.2 Ozone loss at high water vapour mixing ratios

For higher water vapour mixing ratios than 11 ppmv, ozone depletion is simulated (Fig. 3). The ozone loss mechanism generally consists of two steps: a chlorine activation step transferring inactive chlorine (HCl) into active $ClO_x$ followed by catalytic ozone loss processes (Anderson et al., 2012). We analyse both the chlorine activation step and subsequent catalytic ozone loss cycles
potentially occurring in mid-latitudes in the lower stratosphere under enhanced water vapour conditions. Since ozone depletion is larger at high water vapour mixing ratios, conditions with a water vapour mixing ratio of 15 ppmv are chosen here to analyse the chemical ozone loss mechanism. Figure 4 shows an overview of the development of important mixing ratios and reaction rates during the 7-day simulation. Panel a illustrates temperature (black line) and surface area density of liquid particles (blue line).
The first phase of the ozone loss mechanism (dark grey background in Fig. 4) is dominated by the occurrence of heterogeneous




reactions. The most important heterogeneous chlorine activation reaction is R1 (Fig. 4b), which leads to the chlorine activation chain (von Hobe et al., 2011)

$$ClO + NO_2 + M \rightarrow ClONO_2 + M \qquad \text{(R22)}$$

$$ClONO_2 + HCl \rightarrow Cl_2 + HNO_3 \qquad \text{(R1)}$$

$$Cl_2 + h\nu \rightarrow 2\,Cl \qquad \text{(R23)}$$

$$2 \times (Cl + O_3 \rightarrow ClO + O_2) \qquad \text{(R7)}$$

net: $HCl + NO_2 + 2\,O_3 \rightarrow ClO + HNO_3 + 2\,O_2$.

This chlorine activation chain yields a transformation of inactive HCl into active $ClO_x$ as well as of $NO_x$ into $HNO_3$. The ozone loss due to this reaction chain is negligible and no depleting effect on ozone occurs during the first phase (Fig. 4c). In

Fig. 4g, the $NO_x$ mixing ratio is seen to decrease and $HNO_3$ increases due R1. Further, in the first phase the HCl mixing ratio decreases, yielding an increase of $ClO_x$ (Fig. 4f). The delay between HCl reduction and $ClO_x$ formation (Fig. 4f) is caused by the combination of the diurnal cycle and the accumulation of $Cl_2$ and $Cl_2O_2$ during night. Both decreasing $NO_x$ and increasing $ClO_x$ have an impact on ozone during the second phase of the ozone loss mechanism (light grey background in Fig. 4), which is characterized by a decreasing ozone mixing ratio (Fig. 4c). The role of $NO_x$ and $ClO_x$ is discussed in detail in the next sections.

### 3.2.1 Role of $NO_x$

The transformation of $NO_x$-radicals into $HNO_3$ is due to R22 ($ClO+NO_2$) and subsequent the occurrence of the heterogeneous reactions R1 ($ClONO_2 + HCl$) and R2 ($ClONO_2 + H_2O$), which form $HNO_3$. This behaviour was also found in former studies (e.g. Keim et al., 1996; Pitari et al., 2016; Berthet et al., 2017), investigating the impact of volcanic aerosols on stratospheric

ozone chemistry. Dependent on temperature and water vapour content, the $HNO_3$ formed is taken up into condensed particles. In the standard simulation using 15 ppmv $H_2O$, an uptake of 64% is reached on the day with the lowest temperature (197.3 K, 2 Aug 2013), while at higher temperatures (4–7 August 2013) 85% of $HNO_3$ remain in the gas phase. After the transformation of $NO_x$ into $HNO_3$, the $NO_x$ mixing ratio remains low in the second phase of the mechanism (Fig. 4d, light grey region) while the $HNO_3$ mixing ratio (cond.+gas) remains high.

The transformation of $NO_x$ radicals into $HNO_3$, due to the occurrence of heterogeneous reactions at elevated water vapour amounts, affects stratospheric ozone chemistry. In the presence of a high $NO_x$ concentration (as at low water vapour mixing ratios), ozone chemistry is dominated by $NO_x$ radicals (see Sec. 3.1) and the ozone formation in cycle **C4** is determined by the rate of R19 (OH+CO). But if the $NO_x$ concentration is low (as in the second phase of the mechanism), this ozone formation cycle is rate limited by R21 ($NO + HO_2$). For the standard case at 15 ppmv $H_2O$, both rates are shown in Fig. 4e. In the first

phase before $NO_x$ is transferred into $HNO_3$, cycle **C4** is limited by R19 (OH+CO) which peaks on 1 August 2013 with a maximum rate of $1.0 \cdot 10^5 \, \text{cm}^{-3}\text{s}^{-1}$. In the second phase at low $NO_x$ concentrations, cycle **C4** is limited by R21 ($NO + HO_2$) which peaks on 3 August 2013 with a maximum rate of $1.5 \cdot 10^4 \, \text{cm}^{-3}\text{s}^{-1}$. Hence, due to the occurrence of the heterogeneous reaction R1 the net ozone formation decreases by at least $3.5 \cdot 10^4 \, \text{cm}^{-3}\text{s}^{-1}$ from 1 August to 3 August.





**Figure 4.** Reaction rates and mixing ratios important for the ozone loss mechanism in the standard simulation using 15 ppmv $H_2O$. The chlorine activation phase is shaded in dark grey, while the phase of ozone loss has an light grey background. Panel (a) shows the temperature of the trajectory and the liquid surface area density, the ozone mixing ratio is presented in panel (c). Heterogeneous reaction rates are shown in panel (b) as well as the rate of the gas phase reaction $CH_4 + Cl$. Panels (d), mixing ratio of $HNO_3$, $NO_x$ and $ClONO_2$, and (e) are relevant to show the role of $NO_y$ for the ozone loss process. Reaction R19 (OH+CO, panel e) limits ozone formation in cycle **C4** at high $NO_x$ mixing ratios and R21 ($HO_2 + NO$ at lower $NO_x$ concentrations. Panels (f)–(h) illustrate the role of chlorine for ozone loss by showing the mixing ratio of HCl, $ClO_x$ and $ClONO_2$ (panel f), main reaction rates (R4 (ClO+ClO), R8 (ClO+BrO), R10 (ClO $+ HO_2$)) for catalytic ozone loss cycles (panel g) and potential pathways for the OH-radical (R19 (OH+CO), R26 (OH+ClO), R12 (OH+$O_3$)) as possible reaction chains following R10 (ClO+$HO_2$) (panel h).





### 3.2.2 Role of ClO$_x$

In the first phase of the mechanism, chlorine activation yields a transformation of inactive chlorine into active ClO$_x$. Net chlorine activation occurs when the rates of the heterogeneous reactions R1(ClONO$_2$ + HCl), R2 (ClONO$_2$ + H$_2$O) and R3 (HCl+HOCl) exceed the gas phase HCl formation dominated by the reaction

$$Cl + CH_4 \rightarrow HCl + CH_3. \tag{R24}$$

Enhanced ClO$_x$ concentrations induce catalytic ozone loss cycles at low temperatures, as the ClO-Dimer-cycle (**C1**) (Molina et al., 1987), the ClO-BrO-cycle (**C2**) (McElroy et al., 1986) and cycle **C3** (Solomon et al., 1986). Under conditions of low water vapour (stratospheric background) the rate limiting steps of these cycles are the reactions

$$ClO + ClO + M \rightarrow ClOOCl + M, \tag{R4}$$

$$ClO + BrO \rightarrow Br + Cl + O_2 \tag{R8}$$

$$\text{and} \quad ClO + HO_2 \rightarrow HOCl + O_2. \tag{R10}$$

The rates of reactions R4, R8 and R10 increase strongly in the second phase of the mechanism (light grey area in Fig. 4g) and thus catalytic ozone loss cycles can occur. Under the assumed conditions, ozone depletion is mainly driven by reaction pathways following both R8 and R10. The reaction rates peak on August 3 with a value of $7.8 \cdot 10^4\,\mathrm{cm^{-3}s^{-1}}$ for R10 (ClO+HO$_2$), $6.8 \cdot 10^4\,\mathrm{cm^{-3}s^{-1}}$ for R8 (ClO+BrO) and $2.7 \cdot 10^4\,\mathrm{cm^{-3}s^{-1}}$ for R4 (ClO+ClO). In contrast the rate of ozone loss due to the reactions

$$ClO + O(^3P) \rightarrow Cl + O_2 \tag{R25}$$

$$\underline{Cl + O_3 \rightarrow ClO + O_2} \tag{R7}$$

net: $O(^3P) + O_3 \rightarrow 2\,O_2$

are not important here, as the peak values or R25 are only about $0.35 \cdot 10^4\,\mathrm{cm^{-3}s^{-1}}$(not shown).

Additionally the sensitivity of various reaction rates to the water vapour mixing ratio was tested. In Figure 5, the mean reaction rates on 3 August are plotted against the water content assumed during the simulation. Panel (a) shows an acceleration of the ozone loss cycles **C2** (based on R8) and **C3** (based on R10) beginning from a water vapour mixing ratio of 11ppmv. In contrast, the rate determining reaction of **C1** (R4, ClO+ClO) increases at a higher water vapour mixing ratio.

At stratospheric background conditions with a low water vapour mixing ratio, the rate determining step of cycle **C3** is R10 (Solomon et al., 1986; Ward and Rowley, 2016). For the conditions with enhanced water vapour of 15 ppmv in the standard simulation, the rate of R12 (OH+O$_3$) is limiting this cycle (Fig. 4f). An investigation of possible reaction pathways of the OH-radical yields that reactions of OH with CO (R19) and ClO (R26) exhibit a rate similar to the reaction with ozone (R12, Fig. 5b).

$$OH + CO \rightarrow H + CO_2 \tag{R19}$$

$$OH + ClO \rightarrow HO_2 + Cl \tag{R26}$$




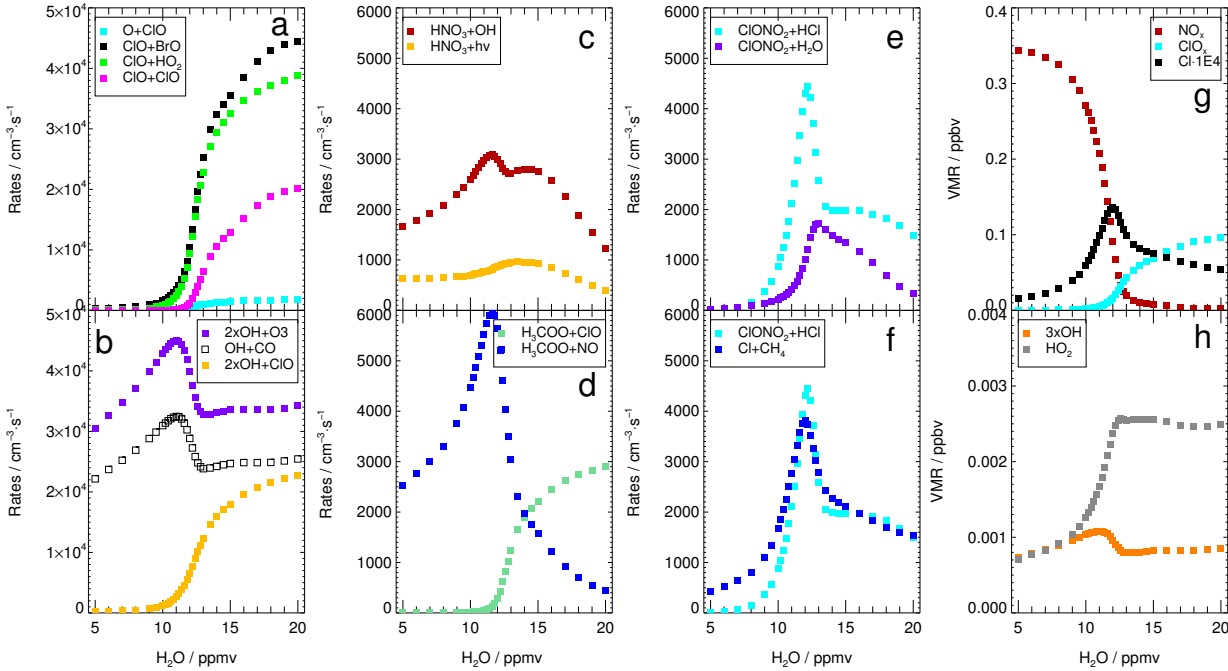

**Figure 5.** Average reaction rates and volume mixing ratios from the standard simulations on 3rd of August dependent on water vapour content. Panel (a) shows the reaction rates of R4 (ClO+ClO), R8 (ClO+BrO), R10 (ClO + HO$_2$) and R25 (ClO+O(3P)) resulting in ozone reduction, panel (b) possible reaction pathways for the OH radical (R19 (OH+CO), R26 (OH+ClO) and R12 (OH+O$_3$)), panel (c) reactions yielding depletion of HNO$_3$ (R31 (HNO$_3$ + OH), R34 (HNO$_3$ + h$\nu$)), panel (d) reactions of the H$_3$COO-radical R27 (H$_3$COO + ClO) and R29 (H$_3$COO + NO), panel (e) important heterogeneous reactions (R1 (ClONO$_2$ + HCl), R2 (ClONO$_2$ + H$_2$O)), and panel (f) the balance between R1 (ClONO$_2$ + HCl) and R24 (CH$_4$ + Cl). Panel (g) shows the mixing ratios of NO$_x$, ClO$_x$ and 10·Cl and panel (h) the mixing ratios of OH and HO$_2$.

Based on these reactions, two further reaction chains affecting ozone can be deduced. In cycle **C5** the OH-radical reacts with CO yielding CO$_2$ and a hydrogen radical, from which HO$_2$ is formed. Subsequently HOCl can be formed via R10 (ClO+HO$_2$) and photolysed in reaction R11. Thus, the net reaction of this pathway is the oxidation of CO to CO$_2$ and the simultanous





destruction of ozone (**C5**).

$$OH + CO \rightarrow H + CO_2 \tag{R19}$$

$$H + O_2 + M \rightarrow HO_2 + M \tag{R20}$$

$$ClO + HO_2 \rightarrow HOCl + O_2 \tag{R10}$$

$$HOCl + h\nu \rightarrow Cl + OH \tag{R11}$$

$$Cl + O_3 \rightarrow ClO + O_2 \tag{R7}$$

$$\text{net: } CO + O_3 + h\nu \rightarrow CO_2 + O_2 \tag{C5}$$

Furthermore, when the OH-radical reacts with ClO, the products are $HO_2$ and Cl and thus another catalytic ozone loss cycle **C6** results.

$$OH + ClO \rightarrow HO_2 + Cl \tag{R26}$$

$$ClO + HO_2 \rightarrow HOCl + O_2 \tag{R10}$$

$$HOCl + h\nu \rightarrow Cl + OH \tag{R11}$$

$$2 \text{ x } (Cl + O_3 \rightarrow ClO + O_2) \tag{R7}$$

$$\text{net: } 2O_3 \rightarrow 3O_2 \tag{C6}$$

In the cycles **C3** and **C6** two ozone molecules are destroyed, while one ozone molecule is destroyed in **C5**. To assess the effectiveness regarding ozone loss of **C3** and **C5** – **C6**, the rate of R19 (limiting **C5**) is compared with two times the rate of R12 (limiting **C3**) and R26 (limiting **C6**). This comparison shows that cycle **C3** is more relevant for ozone loss than **C5** and **C6** (Fig. 5b). Reaction R12 (**C3**) and R19 (**C5**) accelerate with increasing water vapour mixing ratio, due to an increasing formation of OH (Fig. 5h), and peak in the threshold range of $\sim$11 ppmv $H_2O$. When the threshold region is reached, the OH mixing ratio decreases due to a declining $NO_x$ mixing ratio (Fig. 5g) and thus a lower rate of the reaction

$$HO_2 + NO \rightarrow OH + NO_2. \tag{R21}$$

The lower OH concentration results in a decrease of the reaction rates of R12 ($OH+O_3$) and R19 ($OH+CO$) shown in Fig. 5b, since R12 ($OH+O_3$) and R19 ($OH+CO$) are limited through the OH mixing ratio. The limiting step of cycle **C6** (R26) is negligible for low water amounts and starts to increase in the range of the threshold region due to the strong gain of both $ClO_x$ and $HO_2$ (Fig. 5g-h). Accordingly the relevance of **C6** for catalytic ozone destruction increases for higher water vapour mixing ratios.

A requirement for the effectiveness of the ozone loss cycles **C1** – **C3** and **C5** – **C6** is a high mixing ratio of active chlorine ($ClO_x$). In Fig. 4b, the rate of the main HCl-forming reaction R24 ($Cl+CH_4$, dark blue) shows a formation of HCl, which has to be balanced (by HCl destroying reactions) to hold the HCl mixing ratio low and thus $ClO_x$ values high. For conditions of the Antarctic polar night in the lower stratosphere this balance between gas phase HCl-formation and heterogeneous





HCl-destruction can be described through HCl-null-cycles (Müller et al., 2018). In these HCl-null-cycles each HCl formed in reaction R24 is depleted through the heterogeneous reaction R3 (HCl+HOCl). For the formation of HOCl in reaction R10 ($HO_2$+ClO), the generation of $HO_2$ radicals through R27 is essential for Antarctic polar night conditions.

$$H_3COO + ClO \rightarrow HCHO + HO_2 + Cl \tag{R27}$$

5    For the conditions in the mid-latitudes during summer considered here, a higher $NO_x$ mixing ratio prevails than under Antarctic ozone hole conditions due to a lower $HNO_3$ uptake into the condensed particles. As a consequence R1 ($ClONO_2 + HCl$) is mainly responsible for the HCl-loss and hence the pathway **C7** represents a more probable reaction chain to balance the HCl-formation.

$$Cl + CH_4 \rightarrow HCl + CH_3 \tag{R24}$$

$$CH_3 + O_2 \rightarrow H_3COO \tag{R28}$$

$$H_3COO + NO \rightarrow NO_2 + H_3CO \tag{R29}$$

$$H_3CO + O_2 \rightarrow HCHO + HO_2 \tag{R30}$$

$$NO_2 + h\nu \rightarrow NO + O(^3P) \tag{R15}$$

$$HO_2 + ClO \rightarrow HOCl + O_2 \tag{R10}$$

$$HOCl + h\nu \rightarrow OH + Cl \tag{R11}$$

$$OH + HNO_3 \rightarrow H_2O + NO_3 \tag{R31}$$

$$NO_3 + h\nu \rightarrow NO_2 + O(^3P) \tag{R32}$$

$$ClO + NO_2 + M \rightarrow ClONO_2 + M \tag{R22}$$

$$ClONO_2 + HCl \rightarrow HNO_3 + Cl_2 \tag{R1}$$

$$Cl_2 + h\nu \rightarrow 2Cl \tag{R23}$$

$$2 \times (O(^3P) + O_2 + M \rightarrow O_3 + M) \tag{R33}$$

$$\underline{2 \times (Cl + O_3 \rightarrow ClO + O_2)} \tag{R7}$$

$$\text{net: } CH_4 + O_2 \rightarrow HCHO + H_2O \tag{C7}$$

Since the reactions R24, R29, R31 and R1 hold the lowest rates in **C7**, these reactions are essential for constituting **C7**. In R24 HCl is formed and afterwards instantly a methylperoxy radical ($H_3COO$) is formed (R28), which reacts with NO (R29). This reaction yields an H-radical, which is rapidly converted into an OH-radical by formation and photolysis of HOCl (R10 and R11). Through the reaction between the OH-radical and $HNO_3$ (R31) and the subsequent photolysis of $NO_3$ (R32), a $NO_2$-radical is released from $HNO_3$. The photolysis of $HNO_3$

$$HNO_3 + h\nu \rightarrow NO_2 + OH \tag{R34}$$

might be a further option to convert $HNO_3$ into active $NO_x$, but the rate of reaction OH+$HNO_3$ (R31) is more than 2.5 times larger than the rate of the $HNO_3$ photolysis (Fig. 5c). The $NO_2$-radical, which is generated in R32, reacts with ClO (R22)





forming $ClONO_2$, which heterogeneously reacts with HCl (R1). As a consequence the HCl, formed in reaction R24, is mainly destroyed in reaction R1 following pathway **C7**, which yields the oxidation of $CH_4$ as the net reaction.

In cycle **C7** the $H_3COO$-radical reacts with NO (R29). As an alternative the $H_3COO$-radical can also react with ClO (R27). In Fig. 5d the rates of R29 and R27 are compared with each other dependent on the water vapour mixing ratio assumed during

the simulation. For the water vapour mixing ratio of the threshold region, reaction R29 dominates and **C7** mainly balances the HCl-formation and destruction. At higher water vapour mixing ratios (more than 15 ppmv), reaction R27 ($H_3COO+ClO$) becomes more important than R29 ($H_3COO+NO$, Fig. 5c), because of a lower $NO_x$ and a higher $ClO_x$ concentration. The lower $NO_x$ mixing ratio is due to a stronger conversion of $NO_x$ into $HNO_3$ as well as a higher $HNO_3$ uptake into the liquid particles at a higher water vapour mixing ratio. Hence for very high water vapour mixing ratios (here, greater than 15 ppmv), cycle **C8**

mainly balances HCl formation.

$$Cl + CH_4 \rightarrow HCl + CH_3 \tag{R24}$$

$$CH_3 + O_2 \rightarrow H_3COO \tag{R28}$$

$$H_3COO + ClO \rightarrow HCHO + HO_2 + Cl \tag{R27}$$

$$HO_2 + ClO \rightarrow HOCl + O_2 \tag{R10}$$

$$HOCl + h\nu \rightarrow OH + Cl \tag{R11}$$

$$OH + HNO_3 \rightarrow H_2O + NO_3 \tag{R31}$$

$$NO_3 + h\nu \rightarrow NO_2 + O(^3P) \tag{R32}$$

$$ClO + NO_2 + M \rightarrow ClONO_2 + M \tag{R22}$$

$$ClONO_2 + HCl \rightarrow HNO_3 + Cl_2 \tag{R1}$$

$$Cl_2 + h\nu \rightarrow 2Cl \tag{R23}$$

$$O(^3P) + O_2 + M \rightarrow O_3 + M \tag{R35}$$

$$3 \times (Cl + O_3 \rightarrow ClO + O_2) \tag{R7}$$

$$\text{net: } CH_4 + 2O_3 \rightarrow HCHO + H_2O + 2O_2 \tag{C8}$$

The net reaction of **C8** is the oxidation of methane ($CH_4$) into formaldehyde (HCHO) with a simultaneous ozone destruction.

Since the ozone destruction due to the catalytic ozone loss cycles **C2** and **C5** is much faster, the ozone destruction in **C8** is negligible compared to the ozone loss cycles discussed above.

However, in this example the heterogeneous HCl-destruction (R1) does not balance the HCl-formation (R24) (Fig. 4b) due to increasing temperatures (Fig. 4a). Higher temperatures decelerate the heterogeneous HCl-destruction and thus result in the slightly increasing HCl-mixing ratio from 4 August–7 August 2013 (Fig. 4f). Such temperature fluctuations (Fig. 4a) affect the

balance between HCl formation and destruction less at higher water vapour mixing ratios, because the heterogeneous HCl-destruction rate (R1) increases for both low temperatures and high water vapour mixing ratios (see Sec. 4). Thus, regarding the balance between HCl formation and HCl destruction (and hence the balance between chlorine deactivation and chlorine



activation), a high water vapour mixing ratio can compensate a small range of temperature fluctuations. This balance can be described by the cycles **C7** and **C8** and maintains active chlorine levels, which is essential for the ozone loss cycles **C1**–**C3** and **C5**–**C6** to proceed.

## 4    Analysis of chlorine activation

In the previous section we showed that in the temperature range of 197–203 K there is a threshold for water vapour, which has to be exceeded to yield substantial ozone destruction. Here we investigate the sensitivity of this threshold on sulphate content, temperature, $Cl_y$ and $NO_y$ mixing ratio.

### 4.1    Sensitivity of the water vapour threshold

Modifying temperature, sulphate amount or the mixing ratios of $Cl_y$ or $NO_y$ yields a shift of the water vapour threshold.
Figure 6 shows the ozone values reached at the end of the 7-day simulation (final ozone) for a variety of sensitivity cases assuming the standard trajectory of the SEAC[4]RS case (left) and the MACPEX trajectory (right).
The water vapour dependent final ozone values for the standard case are plotted as blue squares (Fig. 6, left). Raising the trajectory temperature by 1 K over the standard case leads to a higher water vapour threshold of 13.5–14.0 ppmv (open red squares), while increasing the sulphate content by a factor of 3 results in a lower threshold region of 9.0–9.5 ppmv (yellow
diamonds). An even larger enhancement of the sulphate content ($10\times$ $H_2SO_4$, magenta diamonds) lowers the water vapour threshold further to a value near 7 ppmv. Reducing the $NO_y$ mixing ratio to 80% of the standard case yields a shift of the threshold to a lower water vapour mixing ratio (green filled triangles), while an equivalent reduction in the $Cl_y$ mixing ratio shifts the threshold region to higher water vapour mixing ratios (black circles). A reduction in $Cl_y$ also results in higher ozone mixing ratios at the end of the simulation.
While the preceding analysis was based on a single SEAC[4]RS trajectory, similar conclusions are reached when the analysis is conducted using a trajectory from the MACPEX campaign. Results of the analysis using the MACPEX initialization and backward trajectory (Fig. 1, black) are shown in the right panel of Fig. 6. A critical water vapour range of 12–13 ppmv is required to produce a reduction of the final ozone value. In the MACPEX case, the final ozone is mostly for all simulations higher than the initial ozone. This is due to the lower $Cl_y$-mixing ratio ($\sim$55 pptv) assumed in the MACPEX case, which yields
lower rates for catalytic ozone loss after chlorine activation occurred. For the MACPEX case, changes in sulphate (Fig. 6, left, yellow diamonds), $Cl_y$ (black circles) or $NO_y$ (green triangles) mixing ratios affect the catalytic ozone destruction water vapour threshold similarly to that observed for the SEAC[4]RS trajectory. This yields the conclusion that in the considered temperature range ($\sim$197–203 K), an ozone reduction occurs after exceeding a water vapour threshold and that this threshold varies with $Cl_y$, $NO_y$, sulphate content and temperature.





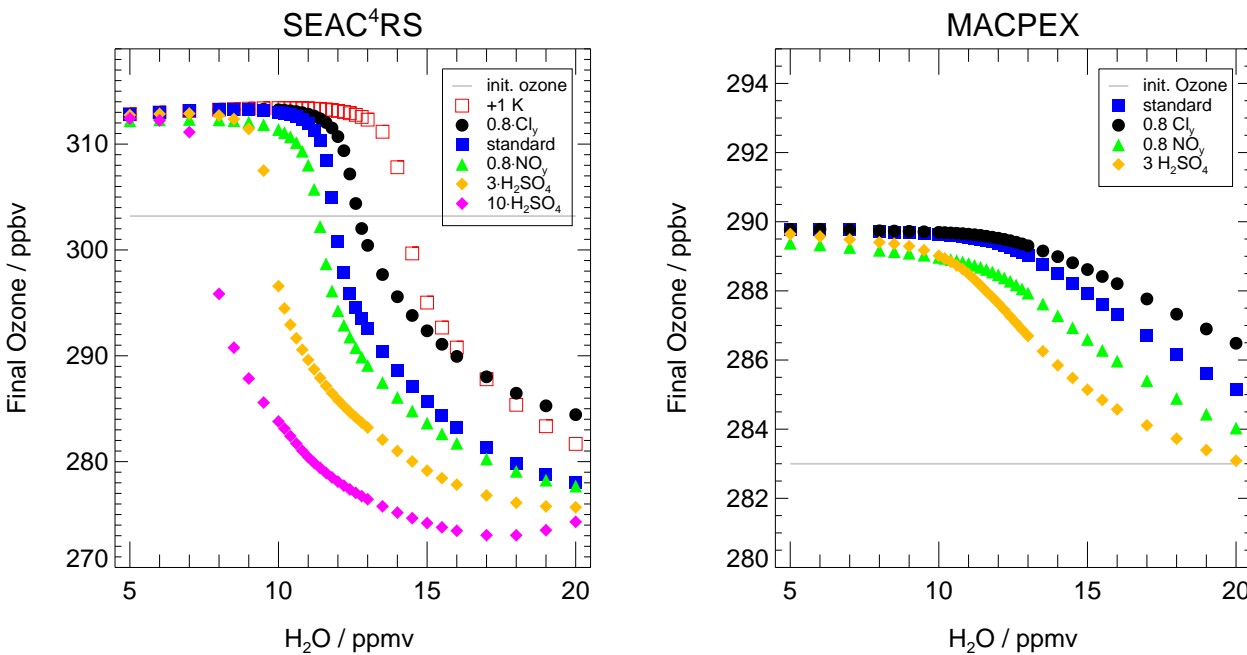

**Figure 6.** Impact of the water vapour content on the ozone mixing ratio (final ozone) reached at the end of the 7-day simulation along the standard trajectory (SEAC$^4$RS, left) and the MACPEX trajectory (right). The standard case is shown in blue and the initial ozone amount is marked by the grey line. An impact on the final ozone mixing ratios is observable after exceeding a critical threshold in water vapour. This threshold changes with a shift in trajectory temperature (+1 K, red), the Cl$_y$ mixing ratio to 0.8 Cl$_y$ (black), the NO$_y$ mixing ratio (0.8 NO$_y$, green) and the sulphate content (3× standard H$_2$SO$_4$, yellow and 10× standard H$_2$SO$_4$, magenta).

## 4.2 Explanation of the water vapour threshold

The sensitivity of the water vapour threshold to Cl$_y$, NO$_y$, sulphate loading and temperature is investigated, focussing on the balance between heterogeneous chlorine activation mainly due to R1 (ClONO$_2$ + HCl) and gas phase chlorine deactivation mainly due to R24 (Cl + CH$_4$). Net chlorine activation takes place when the chlorine activation rate exceeds the chlorine deactivation rate. Reaction R1 is the key reaction in the chlorine activation process. Therefore, in the following, first the dependency of R1 on the water vapour content is analysed in detail. Second, the balance between chlorine activation and deactivation is investigated, also considering the impact of Cl$_y$, NO$_y$ sulphate and temperature on the threshold.





In general the rate of R1 ($ClONO_2 + HCl$) $v_{R1}$ is determined through:

$$v_{R1} = k_{R1} \cdot c_{ClONO_2} \cdot c_{HCl} \tag{1}$$

The concentrations of $ClONO_2$ $c_{ClONO_2}$ and HCl $c_{HCl}$ are associated with the gas phase mixing ratio and the rate constant $k_{R1}$, as a measure of the reactivity of the heterogeneous reaction, depends in this case on the $\gamma$-value $\gamma_{R1}$, the surface area of the liquid particle $A_{liq}$, the temperature $T$ and $c_{HCl}$ (Eq. 2) (Shi et al., 2001).

$$k_{R1} \propto \frac{\gamma_{R1} \cdot A_{liq} \cdot \sqrt{T}}{1 + c_{HCl}} \tag{2}$$

The $\gamma$-value describes the uptake of $ClONO_2$ into liquid particles due to the decomposition of $ClONO_2$ during reaction R1 and is thus a measure of the probability of the occurrence of this heterogeneous reaction (Shi et al., 2001). From Eq. 2 it is obvious that a large surface area $A_{liq}$ and a high $\gamma$-value $\gamma_{R1}$ increase $k_{R1}$ and thus the heterogeneous reaction rate $v_{R1}$.

In Figure 7, the impact of the water vapour content on $\gamma_{R1}$, $A_{liq}$, $k_{R1}$ and the reaction rate $v_{R1}$ is plotted. To avoid the influence of R1 itself on these parameters as much as possible, these parameters are selected for 1 August 2013 at 13:00 UTC. This point in time corresponds to the values after the first chemistry time step during the chemical simulation. The standard case is illustrated in blue squares (Fig.7) and exhibits a strongly increasing gamma value especially for water vapour mixing ratios between 9 and 14 ppmv as well as an almost constant liquid surface area $A_{liq}$ in the same water vapour range. The slight increase in $A_{liq}$ is caused by $HNO_3$ formation in R1 and the subsequent uptake of $HNO_3$ into the condensed particles, especially for high water values. Due to an increasing $\gamma$-value with increasing water vapour, the rate constant $k_{R1}$ increases (Shi et al., 2001) and thus induces a larger reaction rate $v_{R1}$ with an increasing water vapour mixing ratio.

At low water vapour mixing ratios, not only the rate of R1 ($ClONO_2 + HCl$) but also of R24 ($CH_4 + Cl$) increases with an increasing water content (Fig. 5f). An increasing heterogeneous reaction rate (R1) results in both a lower $NO_x$ mixing ratio and more HCl converted into $ClO_x$. A higher $ClO_x$ concentration yields a higher Cl mixing ratio and thus an increase in the rate of R24 ($CH_4 + Cl$). Since both the rates of R1 and R24 increase, no significant net chlorine activation occurs. In the water vapour threshold region, the Cl-mixing ratio peaks (Fig. 5g), because less ClO is converted into Cl through R17 (ClO+NO) due to the decreasing $NO_x$ mixing ratio. The lower Cl mixing ratio reduces the HCl formation in R24 ($CH_4 + Cl$). Hence, the increasing heterogeneous reactivity $k_{R1}$ impedes R24 by reducing the $NO_x$ mixing ratio. In the same way, an increase in $k_{R1}$ yields a higher rate of R1. As a consequence the rate of R1 exceeds the rate of R24 and a net chlorine activation takes place, leading to a reduction of HCl. The decline in both HCl and $NO_x$ yields smaller rates of R1 and R24 at high water amounts and thus the peak of R1 and R24 occurs in the water vapour threshold region (Fig. 5f). Hence, the increasing heterogeneous reactivity ($k_{R1}$) of R1 promotes chlorine activation (due to an increasing rate of R1) and impedes chlorine deactivation (due to a reduction of R24). This yields heterogeneous chlorine activation to exceed gas phase HCl-formation in the water vapour threshold region. The increase in $k_{R1}$ yields a net chlorine activation in the water vapour threshold region by destabilizing the balance between chlorine activation and deactivation.

For an enhanced sulphate content (Fig. 7, yellow diamonds), the particle surface area density (illustrated by $A_{liq}$) is larger, leading to both a stronger increase of the heterogeneous reactivity ($k_{R1}$) and a higher heterogeneous reaction rate than in the





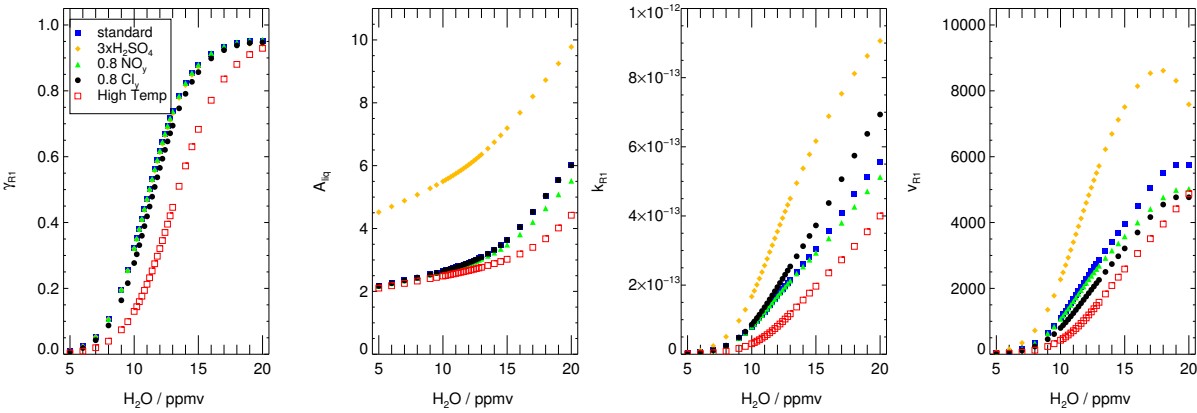

**Figure 7.** Dependence on water vapour of the rate of the the main heterogeneous chlorine activation reaction R1, the rate coefficient ($k_{R1}$), the $\gamma$-value $\gamma_{R1}$ and the liquid surface area density $A_{liq}$. Presented parameters correspond to the values after the first chemistry time step of the box-model simulation. Additionally the impact of an enhanced sulphate content (0.8 ppbv $H_2SO_4$, yellow), reduced $NO_y$ (0.8 $NO_y$, green), reduced $Cl_y$ (0.8 $Cl_y$, black) and enhanced temperatures (red) is shown. The standard case is shown as blue squares.

standard case. Due to this higher heterogeneous reactivity ($k_{R1}$), the chlorine activation rate exceeds the chlorine deactivation at a lower water vapour mixing ratio and the net chlorine activation is reached at a lower water vapour threshold. A shift to higher temperatures (Fig. 7, red) yields almost no change in the surface area density ($A_{liq}$) but a reduced $\gamma$-value and thus a lower heterogeneous reactivity ($k_{R1}$). The reduced reactivity causes the net chlorine activation to occur at a higher water vapour threshold. In contrast, the shift of the threshold for simulations with only 80% of standard $NO_y$ (0.8 $NO_y$, Fig. 7 green) or $Cl_y$ (0.8 $Cl_y$, Fig. 7 black) can not be explained only by an increase in $k_{R1}$. In these cases, further effects on the balance between chlorine activation and chlorine deactivation have to be taken into account. The water vapour threshold in the 0.8 $NO_y$ simulation (green triangles) is shifted to lower water vapour values due to a smaller Cl/ClO-ratio for lower $NO_x$ concentrations. This yields a reduced HCl formation through R24 ($CH_4 + Cl$) than in the standard case and thus impedes chlorine deactivation.





The reduced chlorine deactivation affects the balance between chlorine activation and deactivation in a way that the water vapour threshold region in the 0.8 $NO_y$ case is lower than in the standard case. In the 0.8 $Cl_y$ case (Fig. 7, black), the HCl and $ClONO_2$ mixing ratios are reduced. This leads to a lower chlorine activation rate $v_{R1}$ than in the standard case, despite of the slight higher heterogeneous reactivity ($k_{R1}$), which is due to the inverse dependence of $k_{R1}$ on the HCl concentration (Eq. 2).

The lower dependence of reaction R24 (Cl+CH$_4$) than of R1 (HCl + $ClONO_2$) on the $Cl_y$ mixing ratio would push chlorine deactivation (R24) in the balance between chlorine activation and deactivation and hence shift the water vapour threshold to higher water vapour mixing ratios. Additionally caused by the lower rate of R1 ($ClONO_2 +$ HCl) for reduced $Cl_y$, the $NO_x$ mixing ratio decreases more slowly. This enhances the rate of R24 compared with the standard case as well, because more $NO_x$ yields a higher Cl/ClO-ratio.

In summary, the threshold is determined by the balance between chlorine activation and deactivation and is thus in a certain temperature range especially sensitive to the water dependence of the heterogeneous reactivity ($k_{R1}$) mainly described through the $\gamma$-value $\gamma_{R1}$ and the particle surface $A_{liq}$. These parameters are dependent on the present temperature and sulphate content. However, further parameters shifting this balance, such as the $NO_y$ and $Cl_y$ mixing ratio, have an impact on the water vapour threshold as well.

## 4.3 Temperature dependence

The water vapour threshold, which has to be exceeded for chlorine activation and stratospheric ozone loss to occur, is mainly dependent on the temperature. To illustrate the impact of both temperature and water vapour mixing ratio on stratospheric ozone, the relative ozone change occurring after a 7-day simulation, in which a constant temperature and water vapour concentration and the $Cl_y$ and $NO_y$ of the standard case is assumed, is shown in Fig. 8. In the left panel, ozone change as a function

of temperature and water vapour is plotted for non-enhanced sulphate amounts. In the right panel, the relative ozone change is shown for $10\times$ standard sulphate to estimate a potential impact of sulphate geoengineering on stratospheric ozone. Since mixing of neighbouring air parcels is neglected in the box-model study, the relative ozone change calculated corresponds to the largest possible ozone change for the conditions assumed. A mixing of air is expected to reduce the water vapour mixing ratio during the time period of the 7-day trajectory and hence could stop ozone depletion before the end of the trajectory is reached.

In addition to the relative ozone change, the threshold for chlorine activation is shown as a white line in both panels. When temperature is held constant, this threshold corresponds to the water vapour threshold discussed above. Chlorine activation occurs at higher water mixing ratios and lower temperatures relative to the white line plotted. Here, chlorine is defined to be activated, if the $ClO_x/Cl_y$ ratio exceeds 10%.

For climatological non-enhanced sulphate amounts (Fig. 8, left), the temperature has to fall below 203 K for chlorine activa-

tion to occur, even for high water vapour mixing ratios of 20 ppmv. For the simultaneous presence of high water vapour and low temperatures an ozone loss of 9% (max. 27 ppbv $O_3$) was found. This maximal ozone loss occurs for a range of low temperatures (195–200 K) and enhanced water vapour mixing ratios (10–20 ppmv), because of a similar time until chlorine activation occurs. If the temperatures are higher and water vapour mixing ratios lower than the chlorine activation line, the ozone mixing ratio increases around 3.5% ($\sim$ 10 ppbv $O_3$). At enhanced sulphate conditions (Fig. 8, right) an ozone loss of



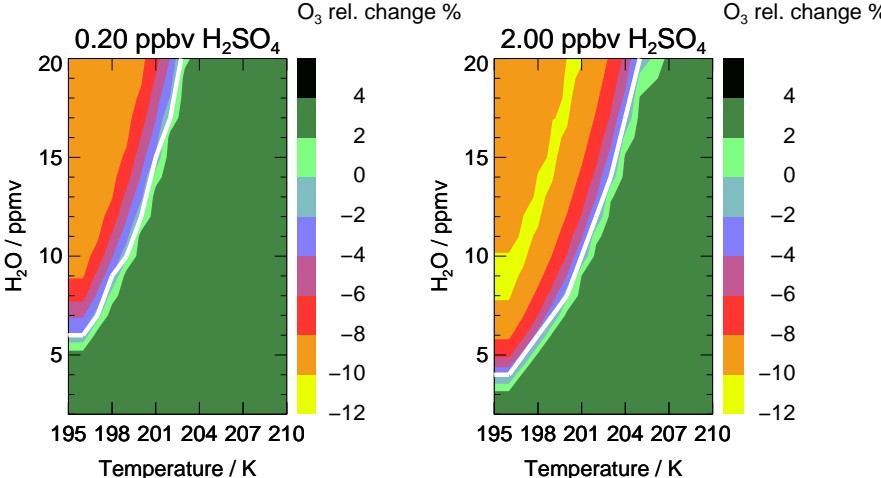

**Figure 8.** Relative ozone change during the 7-day simulation along the standard trajectory dependent on temperature and $H_2O$ ratio for climatological non enhanced (left panel) and enhanced (right panel) sulphate conditions. The white line corresponds to the water and temperature dependent chlorine activation threshold.

max. 10% (30 ppmv $O_3$) occurs for low temperatures and high water vapour mixing ratios. For a water vapour mixing ratio of 20 ppmv the temperature has to fall below 205 K for ozone loss to occur. If the temperatures are very low (195–200 K) and the water vapour is high (10–20 ppmv) ozone loss is slightly reduced. This turnaround occurs, because at a high sulphate loading in combination with high water and low temperatures more HCl is taken up by condensed particles. This leads to less $Cl_y$ in
5  the gas phase and thus lower rates of catalytic ozone loss.

In summary, the combination of low temperatures, enhanced sulphate concentrations and high water vapour mixing ratios promotes an ozone decrease of up to ∼10% (max. −30 ppbv $O_3$) for high water vapour mixing ratios, low temperatures and enhanced sulphate conditions. In comparison to the study of Anderson et al. (2012), the temperatures have to fall below 203 K (here) instead of 205 K (in Anderson et al. (2012)) for non enhanced sulphate conditions and below 205 K instead of 208 K (in
10  Anderson et al. (2012)) for enhanced sulphate conditions and a water vapour mixing ratio of 20 ppmv for chlorine activation and thus ozone loss to occur. Hence, Anderson et al. (2012) found ozone loss in mid-latitudes at high water vapour mixing ratios for temperatures 2 to 3 K higher than in our simulations.





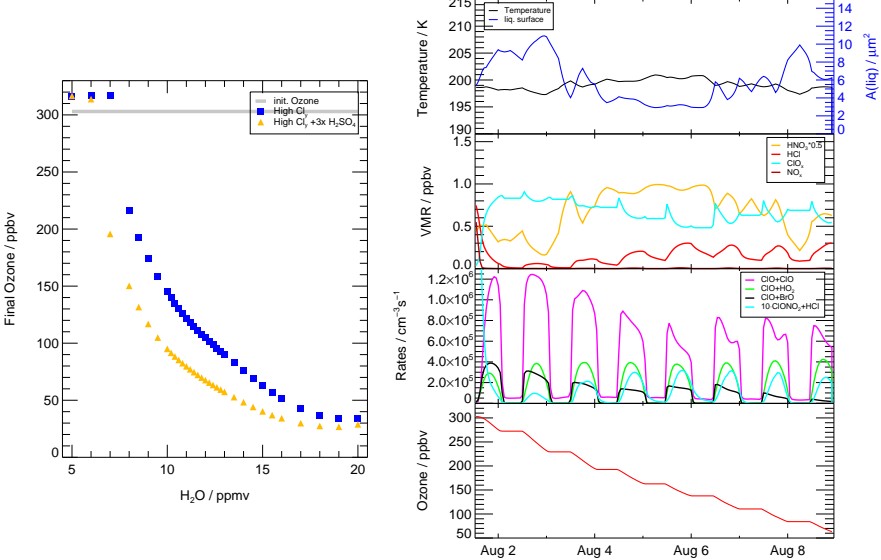

**Figure 9.** Behaviour under "Case of high $Cl_y$" conditions assuming high values for $Cl_y$ and $NO_y$ (see Tab. 1). Left panel shows the dependence of ozone values reached at the end of the 7-day simulation on water vapour for reference (blue) and 3×enhanced sulphate contents (yellow); the grey line corresponds to the initial ozone value. Right panel presents the volume mixing ratio of ozone, temperature, liquid surface density, mixing ratio of $HNO_3$ (scaled by 0.5), HCl, $ClO_x$ and $NO_x$ and reaction rates of reactions essential for chlorine activation and catalytic ozone loss cycles (R1 and R8 (ClO+BrO), R4 (ClO+ClO) and R10 (ClO + $HO_2$)).

## 5 Case studies

Case studies were conducted to illustrate the sensitivities described above on ozone loss and to estimate the impact of realistic conditions and an upper boundary on the ozone loss process. As a kind of worst case study (upper boundary), the "Case of high $Cl_y$" was simulated using $Cl_y$ and $NO_y$ mixing ratios based on the study of Anderson et al. (2012), which uses $Cl_y$

5 and $NO_y$ much larger than inferred from tracer-tracer correlations (Table 1). In the "case based on observations", standard conditions and the measured water vapour mixing ratio of 10.6 ppmv were assumed using both the low sulphate content of the standard case and a slightly enhanced sulphate content, which represents the possible impact of volcanic eruptions or geoengineering conditions. In the "reduced $Br_y$ case", standard conditions with a 50% reduced $Br_y$ mixing ratio were assumed to test uncertainties in current observations of stratospheric bromine burden. Additionally the previously noted standard 7-day

10 trajectory was extended to a 19-day trajectory.



## 5.1 Case of high $Cl_y$

Under conditions of substantially higher initial $Cl_y$ and $NO_y$ mixing ratios (see Tab. 1) than in the standard case used in Anderson et al. (2012), a larger ozone loss up to 265 ppbv during the 7-day simulation is simulated (Fig. 9). Since these high $Cl_y$ conditions have been criticised in other studies (e.g. Schwartz et al., 2013; Homeyer et al., 2014) as being unrealistically

high, they are assumed here as a worst case scenario. Under high chlorine conditions, and for a high water vapour content (more than $\approx 18$ ppmv), an almost complete ozone destruction with an end ozone value of less than 50 ppbv is simulated (Fig. 9, left), which corresponds to parcel ozone loss of 85%. During the 3.5-day simulation in the study of Anderson et al. (2012), an ozone loss of 20% with respect to initial ozone occurs for 18 ppmv $H_2O$. This difference in relative ozone loss for similar conditions here and in the study of Anderson et al. (2012) is caused by a longer assumed ozone destruction period in our simulation.

Assuming the measured water vapour content of 10.6 ppmv for high chlorine conditions would lead to an ozone depletion of 57% during the 7-day simulation. In comparison, in the standard case an ozone loss of 8% is reached when a high water vapour mixing ratio of 20 ppmv is assumed. However, even for the standard trajectory and a high chlorine content, a water vapour amount of 8 ppmv has to be exceeded to yield any ozone reduction. This threshold shifts from 8 ppmv to 7 ppmv for the case where stratospheric sulphate is tripled (Fig. 9, left, yellow triangles).

Comparing the standard case and the high $Cl_y$ case using 15 ppmv water vapour conditions, in the high $Cl_y$ case more inactive chlorine is converted into active $ClO_x$ on the first day of the simulation (Fig. 9, right). This higher $ClO_x$ mixing ratio results in faster catalytic ozone loss cycles with peak values of $3.9 \cdot 10^5 \, cm^{-3} s^{-1}$ for R10 ($ClO + HO_2$), $2.0 \cdot 10^5 \, cm^{-3} s^{-1}$ for R8 ($ClO + BrO$) and $10.9 \cdot 10^5 \, cm^{-3} s^{-1}$ for R4 ($ClO + ClO$) on 3 Aug 2013. Since the $Cl_y$-mixing ratio is much higher than in the standard case, the catalytic ozone loss cycles are dominated by the ClO-Dimer cycle and result in a much larger ozone loss than in the standard

case assuming realistic $Cl_y$ and $NO_y$ mixing ratios.

## 5.2 Case based on observations

The simulation of the case based on observations during the SEAC[4]RS aircraft campaign corresponds to the most realistic case for today's climate. It is identical to that of the standard case but assumes a fixed water vapour mixing ratio of 10.6 ppmv observed on 8 August 2013. Under these conditions, neither heterogeneous chlorine activation due to R1 ($ClONO_2 + HCl$)

nor catalytic ozone loss cycles (e.g. based on $ClO + BrO$) can be observed in the simulation (Fig. 10, left). Instead, ozone is formed due to cycle **C4**. In comparison, the same simulation with 0.6 ppbv gas phase equivalent $H_2SO_4$ instead of 0.2 ppbv was conducted (Fig. 10, right). The enhanced sulphate content yields a larger liquid surface area density and thus an increased heterogeneous reactivity. Hence, reaction R1 occurs in the $3 \times H_2SO_4$ simulation significantly, leading to a slightly increasing $ClO_x$ mixing ratio and a decrease of the $NO_x$ mixing ratio. Both a reduced ozone formation in **C4** (which is at decreased $NO_x$

concentrations limited by R21) and ozone loss cycles (e.g. based on the reaction $ClO + BrO$ or $ClO + HO_2$) can be observed, resulting in a reduction of ozone.

Using initial conditions, the trajectory corresponding to the SEAC[4]RS observations shows ozone loss with sulphate enhanced by a factor of 3. However, we note that this was an unusually cold trajectory. A more common case with higher mean tem-



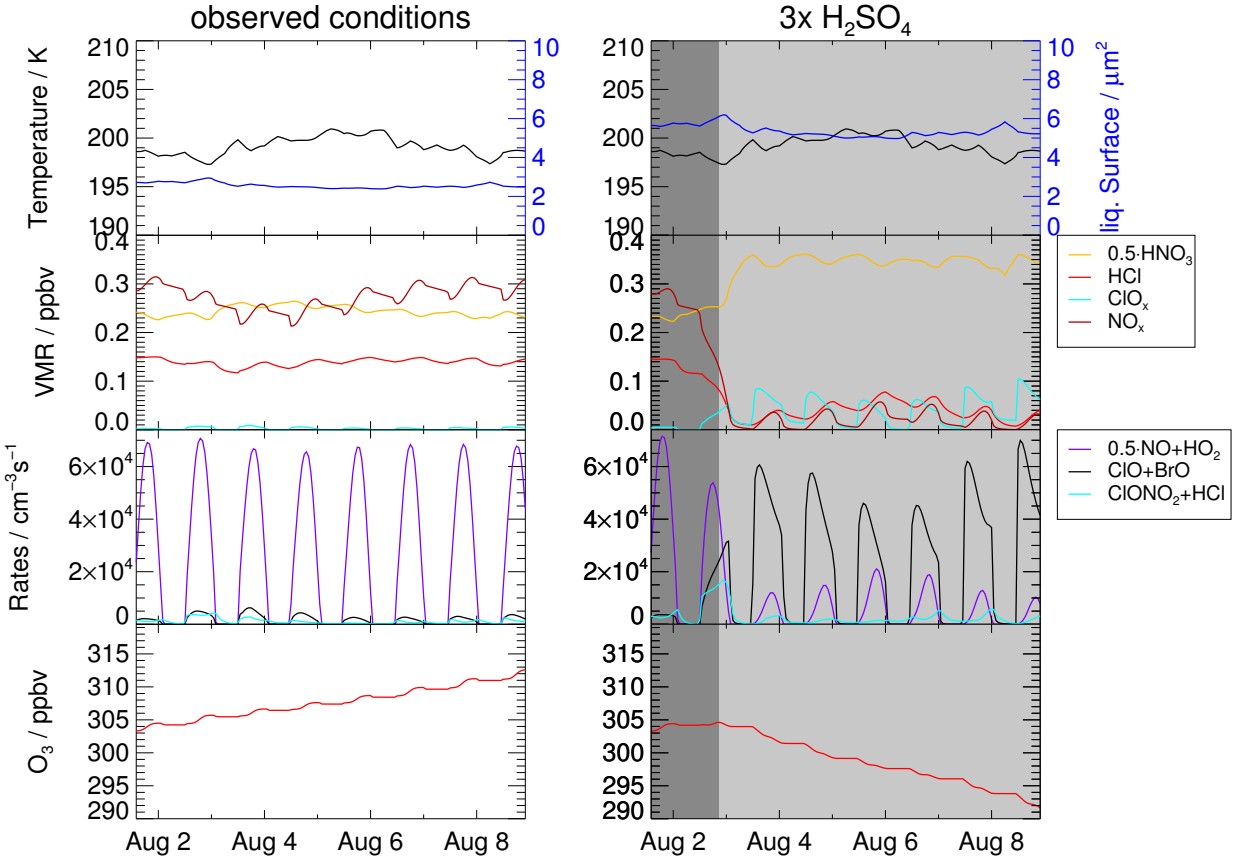

**Figure 10.** Left panels present the temperature, liquid surface area density, ozone mixing ratio, reaction rates of R1 ($ClONO_2 + HCl$, cyan), R8 (ClO+BrO, black) (as an example for ozone loss cycles), R21 ($NO + HO_2$, violet) which limits ozone formation at low $NO_x$ concentrations as well as volume mixing ratios of HCl (red), $ClO_x$ (light blue), $NO_x$ (black) and $HNO_3$ (scaled with 0.5) for conditions of the measurement with 10.6 ppmv $H_2O$ and 0.20 ppbv $H_2SO_4$. The panels on the right show the same quantities, but for enhanced sulphate conditions (0.60 ppbv $H_2SO_4$).

peratures would require a higher sulphate content to enhance the heterogeneous reactivity enough for chlorine activation to occur.

## 5.3 Reduced $Br_y$ Case

The mixing ratio of inorganic bromine ($Br_y$) has a high uncertainty in the lowermost stratosphere due to the influence of very short lived bromine containing substances. For example, during the CONTRAST field campaign (Jan-Feb 2014, western Pacific region), Koenig et al. (2017) observed a $Br_y$ mixing ratio in the lower stratosphere of 5.6–7.3 ppt and the contribution of $Br_y$, which crosses the tropopause, was estimated to be 2.1±2.1 ppt (Wales et al., 2018). Navarro et al. (2017) found somewhat



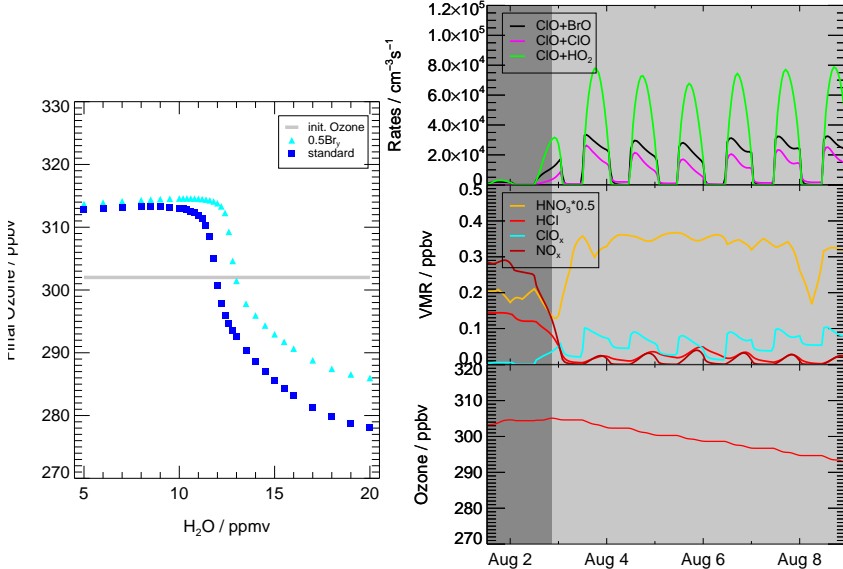

**Figure 11.** Impact of a $Br_y$ reduction on the water vapour threshold and the ozone loss process. Left panel shows the dependence of ozone values reached at the end of the 7-day simulation on water vapour for the standard case (blue) and a simulation assuming the half of $Br_y$ (light blue); the grey line corresponds to the initial ozone value. Right panel presents the volume mixing ratio of ozone, the mixing ratio of $HNO_3$ (scaled by 0.5), HCl, $ClO_x$ and $NO_x$ and reaction rates of reactions essential for chlorine activation and catalytic ozone loss cycles (R1 ($ClONO_2 + HCl$) and R8 (ClO+BrO), R4 (ClO+ClO) and R10 ($ClO + HO_2$)).

different bromine partitioning depending on the ozone, $NO_2$ and $Cl_y$ concentrations, using very short lived bromine species observations in the eastern and western Pacific ocean from the ATTREX campaign. Because our Bry values are not based on measurements for this specific case modeled, we tested the sensitivity to a value that is half of our standard case. The impact of this $Br_y$ reduction is illustrated in Fig. 11 assuming a water vapour mixing ratio of 15 ppmv.

5  Comparing the end ozone value for the $0.5 Br_y$ simulations (Fig. 11, left, light blue triangles) with those of the standard case (blue squares), a higher water vapour threshold and a reduced ozone loss at high water vapour mixing ratios is simulated. The reduction of $Br_y$ yields a slightly longer time period of chlorine activation. At high HCl and low $ClONO_2$ mixing ratios at the start of the simulation, the formation of $ClONO_2$ in R22 (ClO+$NO_2$) is essential for maintaining the chlorine activation reaction R1 ($ClONO_2 + HCl$). Hence, the chlorine activation is dependent on the formation of $ClONO_2$ and thus on the $NO_2/NO$-ratio

10  (von Hobe et al., 2011). A higher $NO_2/NO$-ratio yields a higher rate of R22 and enhances the rate of R1. A reduced $Br_y$-mixing ratio leads to a smaller $NO_2/NO$-ratio due to R18 (BrO+NO) and thus to a lower chlorine activation rate (von Hobe et al., 2011). A reduction of the chlorine activation rate (R1) would change the balance between chlorine activation and chlorine deactivation, which determines the water vapour threshold region. Thus, this reduction would lead to the shift of the water vapour threshold, which is illustrated in Fig. 11 (left panel). With less $Br_y$ the catalytic ozone destruction in the ClO-BrO-cycle





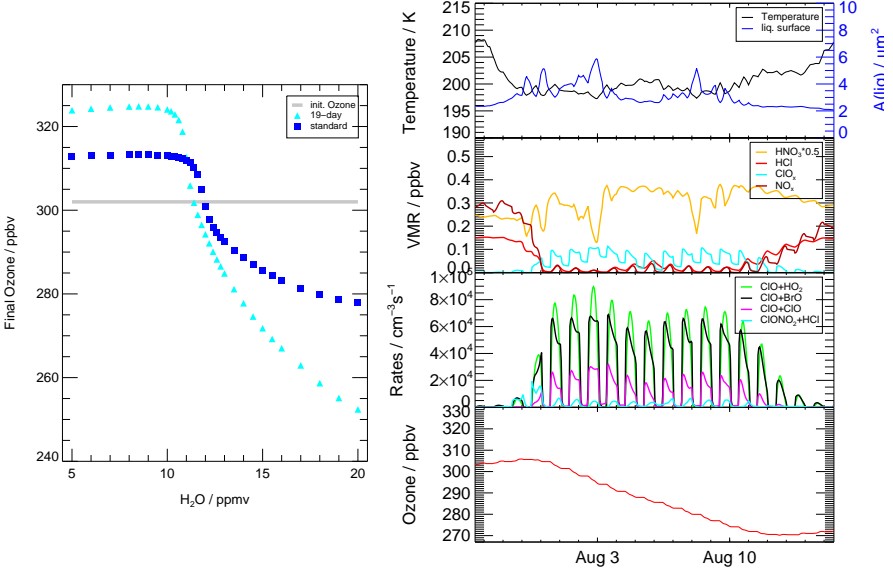

**Figure 12.** Impact of the simulated time period on ozone loss. Left panel shows the dependence of ozone values reached at the end of the simulation on water vapour for the standard case (7-day, blue) and the 19-day simulation (cyan); the grey line corresponds to the initial ozone value. Right panel presents temperature, liquid surface area, the mixing ratio of ozone, $HNO_3$ (scaled by 0.5), HCl, $ClO_x$ and $NO_x$ and reaction rates of reactions essential for chlorine activation and catalytic ozone loss cycles (R1 ($ClONO_2 + HCl$) and R8 (ClO+BrO), R4 (ClO+ClO) and R10 ($ClO + HO_2$)) for the 19-day simulation.

(**C2**) is reduced (Fig. 11, right panel), while the rates of R4 (ClO+ClO) and R10 (ClO+HO$_2$) are similar to those of the standard case (Fig. 4,e). This results in the reduced ozone destruction in the 0.5 $Br_y$ case.

## 5.4 19-day simulation

Since the occurrence of the ozone loss process analysed in this study is strongly dependent on a variety of parameters, the time
5 period over which the ozone loss might occur is very uncertain. The impact of this time period on ozone loss was tested by extending the 7-day trajectory used in the sections above to span the entire period with temperatures low enough to maintain chlorine activation. The temperature development of this trajectory is shown in Fig. 12. On 27 July 2013, the 19-day simulation starts at a temperature of 208 K (Fig. 12, right), decreasing until 29 July 2013 to lower than 200 K. The temperatures remain lower than 201 K until 11 August and increase to over 205 K on 14 August 2013.
10 Assuming a water vapour mixing ratio of 15 ppmv, chlorine activation occurs on 30 July 2013, after the temperatures fall below 200 K (Fig. 12, right). The mixing ratio of $NO_x$ remains low and $ClO_x$ remains high until 11 August, when the heterogeneous reaction rate of R1 ($ClONO_2 + HCl$) decreases due to higher temperatures. For this reason chlorine activation cannot anymore be maintained by cycle **C7**. Thus, the time span holding a $ClO_x$ mixing ratio high enough for the occurrence of catalytic ozone



loss cycles (**C1**–**C3**, **C5**–**C6**) comprises 14 days and ozone destruction stops on 12 August.

Because of the extended time period, the final ozone values using the enhanced water vapour mixing ratios for 19-days (cyan triangles Fig. 12, left panel) are much lower than those of the standard 7-day simulation (blue squares). Additionally, more ozone is formed when using low water vapour concentrations. Comparing the water vapour threshold of the 7-day trajectory (∼11.0–11.6 ppmv) and the 19-day simulation (∼10.6 ppmv), a shift to lower water vapour mixing ratios occurs in the 19-day trajectory. This shift is likely due to an extended time period with a temperature well below 200 K, which allows a chlorine activation to occur even for slightly lower water vapour amounts. Simulations along a trajectory starting on the same day as the 7-day trajectory, but finishing on 15 August, yield the same water vapour threshold as the 7-day simulation (not shown), indicating that the shift in the threshold shown in Fig. 12 is associated with the very cold conditions at the start of the 19-day simulation. Hence, the length of the chosen trajectory has no impact on the water vapour threshold, but does effect the final ozone.

## 6 Discussion

Many uncertainties affect the assessment of the extent of ozone loss that occurs in the lowermost stratosphere at mid-latitudes under elevated water vapour conditions. The number and depth of convective overshooting events as well as the area and duration affected by enhanced water vapour mixing ratios is a subject of recent research (e.g. Homeyer et al., 2014; Smith et al., 2017). The mixing ratio of important trace gases ($O_3, Cl_y, Br_y, NO_y$) in overshooting plumes and the possibility that water vapour mixing ratios high enough for chlorine activation meet temperatures low enough is a matter of debate (e.g. Schwartz et al., 2013; Homeyer et al., 2014). In this study, we examined the sensitivity to different water vapour mixing ratios, temperature, $H_2SO_4$ content, $Cl_y, NO_y, Br_y$ and trajectory duration.

The ozone loss mechanism investigated here requires the occurrence of the heterogeneous reaction R1, which leads to enhanced $ClO_x$ and reduced $NO_x$ mixing ratios and thus maintains effective catalytic ozone loss cycles. Enhanced ClO and reduced NO concentrations were observed by Keim et al. (1996) and Thornton et al. (2007) close to the mid-latitude tropopause under conditions with elevated water vapour and enhanced concentrations of condensation nuclei, such as sulphate particles. These observations were attributed to the occurrence of the heterogeneous reactions R1 ($ClONO_2$+HCl) and R2 ($ClONO_2 + H_2O$, Thornton et al., 2007; Keim et al., 1996). For the temperature and the water vapour range observed in the studies of Keim et al. (1996) (15 ppmv $H_2O$, ∼207 K) and Thornton et al. (2007) (15–22 ppmv $H_2O$, ∼213–215 K), a heterogeneous chlorine activation would not occur in the box-model simulation conducted here, not even in a sensitivity simulation assuming a high sulphate gas phase equivalent of 7.5 ppbv $H_2SO_4$ (not shown). At low temperatures (≲196 K), heterogeneous chlorine activation may occur in the tropical stratosphere (Solomon et al., 2016; von Hobe et al., 2011). Von Hobe et al. (2011) observed enhanced ClO mixing ratios during aircraft campaigns over Australia (SCOUT-O₃, 2005) and Brazil (TROCCINOX, 2005) in combination with low temperatures and the occurrence of cirrus clouds. Analysing the balance between chlorine activation and deactivation von Hobe et al. (2011) showed an increase of the chlorine activation rate (R1) with a higher ClO, BrO and $O_3$ mixing ratio. Thus, once started, R1 accelerates due to higher ClO-mixing ratios subsequently yielding a fast conversion



of $NO_x$ into $HNO_3$ (von Hobe et al., 2011), comparable to the $NO_y$ repartitioning found in the present study. Von Hobe et al. (2011) found a threshold in ozone mixing ratio, which has to be exceeded for chlorine activation to occur. Hence, the water vapour threshold discussed here is expected to depend on the ozone mixing ratio, as well. Furthermore a potential occurrence of ice particles in the lowermost mid-latitude stratosphere (Spang et al., 2015) might affect the water vapour threshold due to

a different heterogeneous reactivity on ice than on liquid particles.

An elevated sulphate content enhances the heterogeneous reaction rate caused by an increased liquid surface. Due to this relation, an impact of stratospheric albedo modification (by applying solar geoengineering) on the ozone loss process proposed by Anderson et al. (2012) is discussed (Dykema et al., 2014). Varying the sulphate content in this study showed that for temperatures and water vapour conditions of the case based on observations, a moderate enhancement of $3\times H_2SO_4$ is sufficient to yield

ozone depletion. Considering the temperature and water vapour dependence of the chlorine activation line (Fig. 8, white line), a $10\times$ enhancement of stratospheric sulphate yields a shift of chlorine activation to slightly lower water vapour mixing ratios and higher temperatures. However, even for enhanced sulphate and a water vapour mixing ratio of 20 ppmv, the temperature has to fall below 205 K for chlorine activation (and hence ozone depletion) to occur at the assumed $Cl_y$ and $NO_y$ conditions of the standard case.

After the chlorine activation step, catalytic ozone loss cycles can occur: the ClO-Dimer cycle (**C1**), the ClO-BrO-cycle (**C2**) and cycles subsequent to R10 ($ClO + HO_2$, **C3**, **C5–C6**). Cycle **C3** is reported to have an impact on stratospheric ozone in mid-latitudes in previous studies (e.g. Johnson et al., 1995; Kovalenko et al., 2007; Ward and Rowley, 2016). Here, **C3** was found to be the dominate cycle based on R10 under standard conditions. Nevertheless, simulating the $0.5 Br_y$ and high $Cl_y$ case has shown that the relevance of **C1** (ClO+ClO) and **C2** (ClO+BrO) depends on assumed initial values of $Cl_y$ and $Br_y$.

Anderson and Clapp (2018) discussed the occurrence of the ClO-Dimer cycle (**C1**) and the ClO-BrO-cycle (**C2**) dependent on water vapour, the $Cl_y$ mixing ratio and temperature. They illustrate a significant increase in the rate of R4 (ClO+ClO) and R8 (ClO+BrO) if the combination of elevated water vapour and low temperatures is sufficient for chlorine activation to occur. If chlorine activation occurs in their model study, a higher $Cl_y$ mixing ratio yields higher catalytic ozone loss rates (R4, R8). Their finding regarding the effect of temperature, water vapour and chlorine on the ozone loss process is consistent with the results

found here. The occurrence of net chlorine activation is determined by temperature and water vapour mixing ratio, while the $Cl_y$ mixing ratio controls how much ozone is destroyed.

A measure for the effect of temperature and water vapour on stratospheric chlorine activation and ozone chemistry is the temperature and water vapour dependent chlorine activation line (Fig. 8, white line). Anderson et al. (2012) estimates lower temperatures than 205 K as necessary for chlorine activation to occur at a water vapour mixing ratio of 20 ppmv and a cli-

matological non enhanced sulphate content. In comparison, assuming standard conditions for $Cl_y$ and $NO_y$ but a constant temperature here, temperatures lower than 203 K are required for similar $H_2O$ and sulphate concentrations. The standard trajectory was chosen here to hold for conditions most likely for chlorine activation based on SEAC[4]RS measurements and at the temperature range of this trajectory, the measured water vapour mixing ratio (10.6 ppmv) is slightly lower than the water vapour threshold. Hence, for all SEAC[4]RS and MACPEX trajectories calculated (not only the shown examples), no trajectory

produced ozone loss. A further requirement for the occurrence of chlorine activation is the maintenance of the conditions,



which yield chlorine activation, during the entire time of chlorine activation. Assuming standard conditions and a water vapour mixing ratio of 20 ppmv, chlorine activation takes 5 hours, but for conditions of the water vapour threshold low temperatures and enhanced water vapour mixing ratios have to be maintained 24–36 hour for chlorine activation and thus an impact on stratospheric ozone chemistry to occur. For the occurrence of ozone depletion, temperatures have also to remain low and water

vapour mixing ratios high after the chlorine activation step.

The maximum ozone depletion at standard conditions occurs here for a water vapour mixing ratio of 20 ppmv. Final ozone at 20 ppmv $H_2O$ in the 7-day simulation is 11% lower than the final ozone reached under atmospheric background conditions of 5 ppmv $H_2O$. For the 19-day simulation at 20 ppmv $H_2O$, the final ozone is 22% reduced compared to the 19-day simulation at 5 ppmv $H_2O$. Anderson and Clapp (2018) calculated a similar ozone reduction of 17% in a 14-day simulation and the same

potential temperature range of 380 K assuming 20 ppmv $H_2O$ and similar $Cl_y$ conditions as used here in the realistic case. In contrast assuming high $Cl_y$ and $NO_y$ of Anderson et al. (2012) in the case of high $Cl_y$ would lead to an ozone loss of 85% (265 ppbv) during the 7-day simulation. This ozone loss would occur in the lower stratosphere. Borrmann et al. (1996, 1997) and Solomon et al. (1997) conducted a study about the impact of cirrus clouds on chlorine activation and ozone chemistry in the mid-latitudes lowermost stratosphere. They found a significant impact of heterogeneous processes occurring on cirrus

clouds for ozone chemistry of the lowermost stratosphere but a minor effect for column ozone. Anderson and Clapp (2018) calculated a fractional loss in the total ozone column of 0.25% assuming a full $Cl_y$ profile in the altitude range of 12–18 km with a constant water vapour mixing ratio of 20 ppmv and the mixing ratio of $Cl_y$ similar to our standard case. However, the simulations assume a constant high water vapour mixing ratio and neglect mixing with the stratospheric background, which is characterized by much lower water vapour mixing ratios and subsequent dilution of convective uplifted air masses. Ozone loss

would only occur in the specific volume of stratospheric air, that is directly affected by the convectively injected additional water. Hence, the ozone loss presented here corresponds to the maximal possible ozone loss for rather realistic convective overshooting conditions.

## 7 Conclusions

We investigated in detail the ozone loss mechanism at mid-latitudes in the lower stratosphere occurring under enhanced water vapour conditions and the sensitivity of this ozone loss mechanism on a variety of conditions. A CLaMS box-model study was conducted including a standard assumption and a variety of sensitivity cases regarding the chemical initialisation, temperatures and the duration of the simulated period. The assumed standard conditions (155.7 pptv $Cl_y$, 728.8 pptv $NO_y$, 197–203 K and an $H_2SO_4$ gas phase equivalent of 0.20 ppbv) were determined based on measurements in an $H_2O$ environment showing strongly

enhanced $H_2O$ values compared to the stratospheric background during the SEAC[4]RS aircraft campaign in Texas 2013.

The ozone loss mechanism consists of two phases: The first step is chlorine activation due to the heterogeneous reaction $ClONO_2 + HCl$ (R1), which yields both an increase of $ClO_x$ and a decrease of $NO_x$. When chlorine is activated, enhanced $ClO_x$ mixing ratios lead to catalytic ozone loss cycles in the second phase of the mechanism. Our findings show that besides



the ClO-Dimer-cycle (**C1**) and ClO-BrO-cycle (**C2**), three ozone loss cycles (**C3**, **C5–C6**) based on the reaction $ClO+HO_2$ (R10) have to be taken into account. The relevance of the cycles **C1–C3** and **C5–C6** for ozone loss depends on the water vapour, $Cl_y$ and $Br_y$ mixing ratios. Reduced $NO_x$ mixing ratios yield a decreasing chemical net ozone formation in cycle **C4**. This reduced ozone formation at high water vapour mixing ratios is in the box-model simulation around 20% as high as the

ozone destruction in catalytic ozone loss cycles. Furthermore a detailed analysis of chemical processes revealed the occurrence of pathways, which maintain high $ClO_x$ and low $NO_x$ mixing ratios after the chlorine activation step but do not reduce ozone, similar to HCl-null-cycles in the lower stratosphere (Müller et al., 2018) in Antarctic early spring.

Focussing on the dependence of chlorine activation on temperature and water vapour mixing ratio, we found that the temperature has to fall below 203 K for chlorine activation to occur at a water vapour mixing ratio of 20 ppmv and $Cl_y$ and $NO_y$ for

our standard case. Testing the water vapour dependence of ozone loss along a realistic trajectory that experienced very low temperatures between 197 and 203 K, we observed a water vapour threshold of 11.0–11.6 ppmv $H_2O$, which has to be exceeded for ozone reduction to occur. For our assumed standard conditions, a maximum ozone loss of 9% (27 ppbv) was calculated for a water vapour mixing ratio of 20 ppmv. In contrast, a simulation assuming the observed conditions (10.6 ppmv $H_2O$) yielded ozone formation; but a tripling of background sulphate gas phase equivalent is sufficient for a slight ozone loss to occur under

these unusually cold conditions for the chosen standard trajectory. Simulating a high $Cl_y$ case assuming initial $Cl_y$ and $NO_y$ based on the study of Anderson et al. (2012) results in both a lower water vapour threshold of 7–8 ppmv and a larger ozone depletion of 85% (265 ppbv) at high water vapour mixing ratios. The model runs described here assume an air parcel moving along the trajectory and does not mix with neighbouring air masses. In the case of water, this would likely reduce the concentration. Because that mixing was neglected, the runs discussed here are likely and extreme case, and the ozone loss modelled

provides an upper bound to the process described.

Considering the duration for which low temperatures and high water vapour mixing ratios have to be maintained to activate chlorine and deplete stratospheric ozone, a chlorine activation time of 24 to 36 hours when the water vapour abundance is near the threshold and of 5 h at 20 ppmv $H_2O$ was calculated. The water vapour threshold shifts strongly with changing temperature and sulphate content as well as with $Cl_y$, $NO_y$ and $Br_y$ mixing ratios. The dependence of the water vapour threshold is

explained here by focussing on the water dependence of the heterogeneous reactivity (R1) and the balance between heterogeneous chlorine activation (R1, $ClONO_2 + HCl$) and gas phase chlorine deactivation (R24, $Cl + CH_4$).

The ozone loss mechanism was investigated here by conducting box-model simulations along a trajectory, which was calculated based on measurements of elevated water vapour. Sensitivity and case studies, which cover a range of uncertainties, illustrate the impact of the $Cl_y$, $NO_y$, $Br_y$ and $H_2O$ mixing ratio, the temperature, the sulphate gas equivalent and the duration

of the simulated period on the ozone loss process. While the water vapour threshold which has to be exceeded for chlorine activation to occur is mainly determined by the temperature, water vapour mixing ratio and sulphate content, the intensity of ozone loss depends on $Cl_y$, $NO_y$, $Br_y$ and the duration of the time period, for which a chlorine activation can be maintained. Our comprehensive sensitivity studies are a basis to assess the impact of enhanced water vapour mixing ratios in the lower mid-latitude stratosphere on ozone under sulphate geoengineering conditions and in a changing climate. However, we did not





simulate ozone depletion for the observed conditions. Further global modelling studies are needed to establish whether the mechanism analysed here is of concern for the future.

*Code and data availability.* The complete SEAC$^4$RS data ar availiable at https://www-air.larc.nasa.gov/cgi-bin/ArcView/seac4rs. The CLaMS box model calculations can be requested from Sabine Robrecht (sa.robrecht@fz-juelich.de).

## 5 Appendix A: Tracer-Tracer Correlations

The mixing ratios of $Cl_y$ and $NO_y$ were initialized based on stratospheric tracer-tracer correlations from Grooß et al. (2014). $Cl_y$ and $NO_y$ were initialized based on a $CH_4$ measurement during the SEAC$^4$RS aircraft campaign. Initial $Cl_y$ was calculated using the tracer-tracer correlation (Grooß et al., 2014)

$$[Cl_y] = 2.510 + 3.517 \cdot [CH_4] - 3.741 \cdot [CH_4]^2 + 0.4841 \cdot [CH_4]^3 + 0.03042 \cdot [CH_4]^4. \tag{A1}$$

The volume mixing ratio of $Cl_y$ ($[Cl_y]$) is here in ppt and the mixing ratio of methane ($[CH_4]$) in ppm.

To determine $NO_y$ based on the $CH_4$ measurement, first N2O was calculated through

$$[N_2O] = -124.9 + 311.9 \cdot [CH_4] - 158.1 \cdot [CH_4]^2 + 146.6 \cdot [CH_4]^3 - 43.92 \cdot [CH_4]^4 \tag{A2}$$

assuming $[N_2O]$ in ppbv and $[CH_4]$ in ppmv (Grooß et al., 2002). Subsequently $NO_y$ (in ppt) was calculated in a correlation with N2O.

$$[NO_y] = 11.57 + 0.1235 \cdot [N_2O] - 1.013 \cdot 10^{-3} \cdot [N_2O]^2 + 1.984 \cdot 10^{-6} \cdot [N_2O]^3 - 1.119 \cdot 10^{-9} \cdot [N_2O]^4 \tag{A3}$$

In the MACPEX case $NO_y$ and $Cl_y$ were initialized based on N2O measurements. $NO_y$ was calculated using correlation A3. $Cl_y$ was calculated using A1. Therefore first $CH_4$ (im ppmv) had to be calculated based on a correlation with N2O (in ppbv) (Grooß et al., 2014).

$$[CH_4] = 0.1917 + 0.01333 \cdot [N_2O] - 8.239 \cdot 10^{-5} \cdot [N_2O]^2 + 2.840 \cdot 10^{-7} \cdot [N_2O]^3 - 3.376 \cdot 10^{-10} \cdot [N_2O]^4 \tag{A4}$$

*Competing interests.* The authors declare that they have no conflict of interest.

*Acknowledgements.* Our activities were funded by the German Science Foundation (Deutsche Forschungsgemeinschaft, DFG) under the DFG project CE-O$_3$ in the context of the Priority Program Climate Engineering: Risks, Challenges, Opportunities? (SPP 1689; VO 1276/4-1). We thank the European Centre for Medium-Range Weather Forecasts (ECMWF) for providing ERA-Interim data. We thank the group of Steven Wofsy (Havard University, Department Earth and Planetary Science, Cambridge, MA USA) and Jessica Smith for providing their 25 data measured during the SEAC$^4$RS aircraft campaign.



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
