# Peer review of "Mechanism of ozone loss under enhanced water vapour conditions in the mid-latitude lower stratosphere in summer"

_Atmospheric Chemistry and Physics, 2018_

## Referee Comment (RC1) · Anonymous Referee #1 · 9 Jan 2019

Review of the paper: Mechanism of ozone loss under enhanced water vapour conditions in the mid-latitude lower stratosphere in summer, by S. Robrecht et al., Atmos. Chem. Phys. Discuss., https://doi.org/10.5194/acp-2018-1193, 2018.

This paper presents a detailed chemical study for a potentially significant ozone depletion in the lowermost stratosphere, using a chemical box-model, with 7-day and 19-day back-trajectory analysis. The study is conducted under conditions of low temperatures (<205 K) and an elevated water vapour mixing ratio, up to 20 ppmv (resulting from convective overshooting events, rather frequent for summertime mid-latitude conditions). These convective events can transport ice crystals into the lowermost stratosphere,

where the ice evaporates leading to a local water vapour increase. The sensitivity to high Cly mixing ratios is also addressed. The authors analyze with plenty of details the catalytic chemical cycles involving ClOx, NOx, HOx and leading to a perturbed balance of ozone production and destruction taking place in the lowermost stratosphere. The study takes inspiration from previous published works (Anderson et al., 2012; Anderson and Clapp, 2018); the authors conclude that the combined effects of temperature, water vapour and chlorine on the ozone loss process are consistent in their study with respect to these previous ones.

I think that this study may help clarify important points regarding the ozone sensitivity to elevated water vapour conditions in the lowermost stratosphere and for this reason I recommend publication on ACP. In my opinion, a few improvements could be made in the manuscript, mainly for completeness and for improving readability.

Specific points

(1) Chemical cycles are essentially those leading to polar ozone depletion and widely described in previous literature. For this reason, I suggest moving large part of chemical details from the main text to a specific Appendix or in supplementary material. In particular, section 3.2.2 is in my opinion way too detailed and should be simplified focusing of the evidences of Fig. 5 (which is clear and exhaustive).

(2) The authors clearly state that their box-model ignores mixing with outside air poorer of water vapour, so that their calculated ozone losses should be interpreted as an extreme case. They also suggest a possible use of their findings in global modelling of the atmosphere for future experiment of sulphate geoengineering under changing climate conditions, or in case of major volcanic eruptions. It should be mentioned that in these cases the large scale latitudinal mixing of atmospheric tracers in the lower branch of the Brewer-Dobson circulation could be significantly affected in sulphate-perturbed conditions due to geoengineering or major tropical eruptions, both leading to a different level of isolation of the tropical pipe with the mid-latitudes. Eddy heat

fluxes could also be perturbed, thus affecting mid-latitude temperatures in the lower stratosphere. Recent works have addressed these specific points (e.g., Visioni et al., 2017).

Visioni, D., et al.: Sulphate Geoengineering Impact on Methane Transport and Lifetime: Results from the Geoengineering Model Intercomparison Project (GeoMIP), Atmos. Chem. Phys., 17, 11209-11226, doi:10.5194/acp-17-11209-2017, 2017.

Minor points

Page 25, lines 6-8: the sentence sounds odd and should be modified: "In summary, the combination of low temperatures, enhanced sulphate concentrations and high water vapour mixing ratios promotes an ozone decrease of up to ∼10% (max. -30 ppbv O3) for high water vapour mixing ratios, low temperatures and enhanced sulphate conditions."

---

## Referee Comment (RC2) · Anonymous Referee #2 · 9 Jan 2019

General comments:

The paper by Robrecht et al. investigates a potential ozone loss mechanism in the mid-latitude lower stratosphere under enhanced water vapour conditions and low temperatures, initiated by heterogeneous chlorine activation on liquid aerosol particles. For that purpose the authors conducted box model simulations along 7(19)-days backward trajectories with the CLamS model. Besides a detailed chemical analysis of their standard case, they investigated the sensitivity of the proposed ozone loss mechanism to water vapour, mixing ratios of Cly, NOy, and Bry, sulphate content and temperature.

Overall, I have no doubts that the applied methods and presented findings are valid

and of wider interest for the scientific community, and therefore, I recommend this paper for publication in ACP. However, the paper is rather lengthy and the presentation of so many different case studies and sensitivity simulations does not increase the readability of the manuscript. Below I tried to make some suggestions for clarifications. Furthermore, I would like to encourage the authors to address or discuss the overall importance of the discussed ozone depletion mechanism. At the very end it is written that for observed conditions no ozone depletion was simulated, but it would be great to put the presented results more into a global and climatological perspective.

Specific comments:

- Abstract: It is a bit weird that the abstract does not clearly mention that the discussed ozone loss mechanism is initiated by heterogeneous chlorine activation on binary (ternary?) solution droplets. There are only "hints" in this direction, namely the impact of volcanic eruptions, geoengineering etc.

- Introduction: Again, it would be good to clearly state that the ozone loss process proposed by Anderson et al. involves sulphate aerosol particles. This would put the subsequent description of heterogeneous chlorine activation and catalytic ozone loss cycles much more into perspective.

- P2, l 15: Is the ClOx definition used here not a bit odd? Why is Cl2 not included? This would also facilitate the understanding of Fig.4f, because then it would be clear where the chlorine released from HCl during the first phase did go. Alternatively, one could also show Cl2 in Fig. 4.

- P3, l17/18: By how much has water vapour to be enhanced to allow Cl activation at higher temperatures? Can you provide a number?

- Model set-up: Why do you neglect NAT and ice in this study? NAT is probably not an issue, but how about ice? Especially since you mention a potential impact of ice particles on the water vapour threshold for Cl activation on liquid aerosol particles due

to different heterogeneous reactivities (p32, l3-5).

- Model set-up: Could provide a short description of the treatment of liquid aerosol particles in your model? In Table 1 you provide the gas phase equivalent $H_2SO_4$ mixing ratio and Fig. 4 ff show the surface area density, but some more information would be great. For example, which $H_2SO_4$ wt% in the liquid aerosol particles is reached under the assumed high water vapour/low temperature conditions?

- P5, l32/33: What was the overall background $H_2O$ mixing ratio for the calculated trajectories? And temperature? By how much are the selected trajectories colder?

- Sect. 2.3 Initialization: In Table 1 it is mentioned that the ClOx species were initialized with 0. Ok, in the mid-latitude lower stratosphere most chlorine is deactivated, but I am wondering how your initial phase in Fig. 4 would look like, if the trajectories had started with non-zero ClOx? Maybe you could add a short discussion.

- Sect. 2.3.3: I do not really see the point of discussing the MACPEX case. In my view it is just another special case that could be covered by the sensitivity simulations. For the sake of clarity, I would suggest to leave it away.

- Fig. 2: What is the rationale behind the chosen $H_2O$ mixing ratios of 5 and 15 ppmv? This choice seems a bit arbitrary to me. Why do you not use the 10.6 ppmv $H_2O$ of your standard case?

- P10, discussion of Fig. 3: In line 3-5 the water vapour threshold is defined as $H_2O$ mixing ratio "at which the end ozone value clearly falls below the end ozone that is reached for low water vapour amounts." What is meant by "clearly"? That sounds a bit vague. Can you be a bit more quantitative? Furthermore, it is stated that "by 12 ppmv of water vapour, the system is clearly in an ozone destruction regime." But is 12 ppmv not just the water vapour mixing ratio at which the transition between ozone production to ozone destruction occurs? At lower $H_2O$ values, end ozone is higher than initial ozone.

- Fig. 4b: Where does the first sharp peak in the light blue line (ClONO2+HCl) come from?

- P13, l20ff: Shouldn't this sentence read: "Dependent on temperature and water vapour content, the HNO3 formed remains in the condensed phase."? And further down: "... 64% of the HNO3 remains in the condensed phase on the day with the lowest temperature, while at higher temperatures... 85% of HNO3 are released to the gas phase...". Is the HNO3 shown in Fig. 4d gas-phase only or total HNO3?

- P18/19 cycles C7 and C8: I have a hard time to follow the construction of these reaction cycles. My impression is that the authors combined different reactions until they ended up with the intended net reaction. For example: In R29 NO2 is formed, which in the next step photolyzes (R15). Lower down another NO2 is formed (R32), but this time it reacts with ClO to ClONO2 (R22).

- P20, l 14-16: I think it would be helpful to mention the reason for the lower H2O threshold under enhanced sulphate conditions, namely the increased SAD. In general, the presentation of the model results would benefit from some more explanation of the underlying processes. This holds also for Sect. 4.2 and the sensitivity of the uptake coefficient to H2O or temperature. By the way, why is the 10xH2SO4 case not shown in Fig. 7?

- P20, discussion of Fig. 6b: I do not understand the statement that similar conclusions as for the SEAC4RS trajectory hold for the MACPEX trajectory. For almost all cases final O3 is higher than initial O3. And why is there only a subset of sensitivities shown for the MACPEX trajectory? Why not also +1K and 10xH2SO4?

- Fig. 6: What is the rationale for the 3xH2SO4 case?

- P25, l3-5: I do not fully understand this argumentation. Do you mean that there is no longer enough ClONO2 available to react with the HCl taken up by the condensed particles and that therefore the enhanced HCl uptake does not lead to further chlorine

activation and ozone loss?

- Fig. 8: This figure nicely shows the different ozone regimes as a function of temperature and H2O for two different sulphate conditions. Would it possible to provide such figure also for Cly, Bry, NOy sensitivities? I am aware that this is a multi-dimensional problem, but I think it would be really helpful to get an overview under which conditions ozone formation and ozone depletion occurs. This would also help to get a better understand of the importance of this mechanism on larger scales.

- Sect. 5: As mentioned above I think the case studies are too much and do not help to provide further insights into the main chemical mechanism. It's rather confusing to distinguish the various atmospheric compositions assumed for the individual model simulations. In particular the 19-days case seems trivial to me: longer time under cold and humid conditions, more ozone loss.

Technical corrections:

- P3, l19: hypothesis hat -> hypothesis that

- P4, l8: ans -> and

- P4, l20: HNO3+2N2O5

- P8, l3: N2O, 2 should be subscript

- P9, l28: ClOx, x should be subscript

- P16, l3: simultanous -> simultaneous

- P17, l29: HCl-formating -> HCl-forming

- P34, l19: and extreme -> an extreme

- Units: The text is a mixture of ppb/ppt and ppbv/pptv. I assume it is always volume mixing ratio?

- Chemical reactions: It would be nice to mark heterogenous reactions as such.

- Table 1: I would suggest to add Cly and NOy for completeness. And are 0.6 ppbv and 2.0 ppbv H2SO4 not also sensitivity simulations?

- Fig 1. It would be nice to mark the time of measurement by a red square or a vertical line also to the left panels. And maybe the tropopause altitude at the location of the air parcel could also be added to the left panel.

- Fig. 4, 9, 10, 11, 12: Which time of day do the x-axis tick marks refer to? One can infer night-/day-time from the reactions with OH, but it would be helpful to provide this information, e.g., in the caption.

- Fig. 7: units (y-axis) are missing
* * *

---

## Author Comment (AC1) · 14 Mar 2019

**Author Comment to Referee #1**

**ACP Discussions doi: 10.5194/acp-2018-1193**
**(Editor - Martin Dameris)**

**'Mechanism of ozone loss under enhanced water vapour conditions in the mid-latitude lower stratosphere in summer'**
* * *
We thank Referee #1 for guidance on how to revise our paper. Following the reviewers advice we shortend the description of chemical processes and refer to a potential change of stratospheric dynamics due to an enhancement of the stratospheric sulphur abundance. Our reply to the reviewer comments is listed in detail below. Questions and comments of the referee are shown in italics. Passages from the revised version of the manuscript are shown in blue.

*This paper presents a detailed chemical study for a potentially significant ozone depletion in the lowermost stratosphere, using a chemical box-model, with 7-day and 19-day back-trajectory analysis. The study is conducted under conditions of low temperatures (<205 K) and an elevated water vapour mixing ratio, up to 20 ppmv (resulting from convective overshooting events, rather frequent for summertime mid-latitude conditions). These convective events can transport ice crystals into the lowermost stratosphere, where the ice evaporates leading to a local water vapour increase. The sensitivity to high Cly mixing ratios is also addressed. The authors analyze with plenty of details the catalytic chemical cycles involving ClOx, NOx, HOx and leading to a perturbed balance of ozone production and destruction taking place in the lowermost stratosphere. The study takes inspiration from previous published works (Anderson et al., 2012; Anderson and Clapp, 2018); the authors conclude that the combined effects of temperature, water vapour and chlorine on the ozone loss process are consistent in their study with respect to these previous ones. I think that this study may help clarify important points regarding the ozone sensitivity to elevated water vapour conditions in the lowermost stratosphere and for this reason I recommend publication on ACP. In my opinion, a few improvements could be made in the manuscript, mainly for completeness and for improving readability.*

**Specific Comments**

*(1) Chemical cycles are essentially those leading to polar ozone depletion and widely described in previous literature. For this reason, I suggest moving large part of chemical details from the main text to a specific Appendix or in supplementary material. In particular, section 3.2.2 is in my opinion way too detailed and should be simplified focusing of the evidences of Fig. 5 (which is clear and exhaustive).*

As proposed, we decided to shorten following aspects to focus on the analysis of lowermost stratospheric ozone chemistry occurring under enhanced water vapour conditions:

- We removed the chemical formulation of the ClO-Dimer-Cycle and the ClO-BrO-cycle in the description of ozone loss cycles leading to polar ozone loss and moved the formulation of cycle C3 from the introduction to section 3.2.2.

- The description of ozone formation at low water vapour mixing ratios is shortened by removing the formulation of ozone chemistry and NO reactions in Section 3.1.

- The detailed explanations of the water vapour dependence of R12 ($OH+O_3$) and R19 ($OH+CO$) in Section 3.2.2 (P.17, l.18-26, ACPD version of the manuscript) is removed.

- The description of the maintenance of activated chlorine is simplified by moving the chemical formulation of cycle C7 and C8 to the appendix. They are substituted by a new scheme (see Fig. 1 of this reply), which illustrates the important relations for the maintenance of elevated chlorine.

*(2) The authors clearly state that their box-model ignores mixing with outside air poorer of water vapour, so that their calculated ozone losses should be interpreted as an extreme case. They also suggest a possible use of their findings in global modelling of the atmosphere for future experiment of sulphate geoengineering under changing climate conditions, or in case of major volcanic eruptions. It should be mentioned that in these cases the large scale*

[Figure]

Figure 1: Scheme to illustrate the balance between chlorine activation and chlorine deactivation (blue, right) and $NO_x$ activation and deactivation (green, left). The heterogeneous reaction $ClONO_2$+HCl (R1) links both cycles. Additional reaction pathways balancing radicals are shown in light colour.

*latitudinal mixing of atmospheric tracers in the lower branch of the Brewer-Dobson circulation could be significantly affected in sulphate-perturbed conditions due to geoengineering or major tropical eruptions, both leading to a different level of isolation of the tropical pipe with the mid-latitudes. Eddy heat fluxes could also be perturbed, thus affecting mid-latitude temperatures in the lower stratosphere. Recent works have addressed these specific points (e.g., Visioni et al.,2017).*

*Visioni, D., et al. : Sulphate Geoengineering Impact on Methane Transport and Lifetime: Results from the Geoengineering Model Intercomparison Project (GeoMIP), Atmos. Chem. Phys., 17, 11209-11226, doi:10.5194/acp-17-11209-2017, 2017.*

We mention the effect of sulphate geoengineering on lowermost stratospheric dynamics in Section 6 (P.27, l.27–31, revised version of the manuscript). We added the following paragraph.

Applying solar geoengineering would also affect the temperature in the lower-most stratosphere by perturbing the Eddy-heat fluxes and would change the lower stratospheric dynamic (Visioni et al., 2017). It would affect large scale latitudinal mixing of atmospheric tracers in the lower branch of the Brewer-Dobson-Circulation leading to a different level of isolation of the tropical pipe with mid-latitudes and would result in a different chemical composition of the lower mid-latitude stratosphere.

**References**

Visioni, D., Pitari, G., Aquila, V., Tilmes, S., Cionni, I., Di Genova, G., and Mancini, E.: Sulfate geoengineering impact on methane transport and lifetime: results from the Geoengineering Model Intercomparison Project (GeoMIP), Atmos. Chem. Phys., 17, 11 209–11 226, doi: 10.5194/acp-17-11209-2017, 2017.

---

## Author Comment (AC2) · 14 Mar 2019

**Author Comment to Referee #2**

**ACP Discussions doi: 10.5194/acp-2018-1193**
**(Editor - Martin Dameris)**
**'Mechanism of ozone loss under enhanced water vapour conditions in the mid-latitude lower stratosphere in summer'**
* * *
We thank Referee #2 for comprehensive guidance on how to revise our paper. Following the reviewers advice we shortened parts of the paper to improve the readability. Our reply to the reviewer comments is listed in detail below. Questions and comments of the referee are shown in italics. Passages from the revised version of the manuscript are shown in blue.

*General comments: The paper by Robrecht et al. investigates a potential ozone loss mechanism in the mid-latitude lower stratosphere under enhanced water vapour conditions and low temperatures, initiated by heterogeneous chlorine activation on liquid aerosol particles. For that purpose the authors conducted box model simulations along 7(19)-days backward trajectories with the CLaMS model. Besides a detailed chemical analysis of their standard case, they investigated the sensitivity of the proposed ozone loss mechanism to water vapour, mixing ratios of Cly, NOy, and Bry, sulphate content and temperature. Overall, I have no doubts that the applied methods and presented findings are valid and of wider interest for the scientific community, and therefore, I recommend this paper for publication in ACP. However, the paper is rather lengthy and the presentation of so many different case studies and sensitivity simulations does not increase the readability of the manuscript. Below I tried to make some suggestions for clarifications. Furthermore, I would like to encourage the authors to address or discuss the overall importance of the discussed ozone depletion mechanism. At the very end it is written that for observed conditions no ozone depletion was simulated, but it would be great to put the presented results more into a global and climatological perspective*

**Specific Comments**

*- Abstract: It is a bit weird that the abstract does not clearly mention that the discussed ozone loss mechanism is initiated by heterogeneous chlorine activation on binary (ternary?) solution droplets. There are only hints in this direction, namely the impact of volcanic eruptions, geoengineering etc.*

Following the reviewer's advice, we mention chlorine activation as initial step for the ozone destruction process in the abstract.

The associated potential ozone loss process requires low temperatures and an elevated water vapour mixing ratio. Since this process is initiated by heterogeneous chlorine activation on liquid aerosols, an increase in sulphate aerosol surface area due to a volcanic eruption or geoengineering could increase its likelihood of occurrence.

*Introduction: Again, it would be good to clearly state that the ozone loss process proposed by Anderson et al. involves sulphate aerosol particles. This would put the subsequent description of heterogeneous chlorine activation and catalytic ozone loss cycles much more into perspective.*

We revised the first part of the introduction to mention that the mechanism proposed by Anderson et al. involves sulphate aerosols and to make clear, that it is initiated by heterogeneous chlorine activation.

Anderson et al. (2012) proposed a potential ozone depletion in the mid-latitude stratosphere in summer on liquid sulphate aerosols under conditions of enhanced water vapour and low temperatures. They proposed this chemical ozone loss to be initiated through heterogeneous chlorine activation and to be driven by catalytic ozone loss cycles related to ozone loss known from polar regions in early spring (e.g. Grooß et al., 2011; Solomon, 1999; Vogel et al., 2011).

*P2, l15: Is the ClOx definition used here not a bit odd? Why is Cl2 not included? This would also facilitate the understanding of Fig.4f, because then it would be clear where the chlorine released from HCl during the first phase did go. Alternatively, one could also show $Cl_2$ in Fig. 4.*

Many thanks for this comment. We added $Cl_2$ to the definition of $ClO_x$ and adapted Fig. 4. The sentence discussing the delay between HCl reduction and $ClO_x$ formation (P. 13 l. 11-12 in the ACPD version of this manuscript) is removed.

*- P3, l17/18: By how much has water vapour to be enhanced to allow Cl activation at higher temperatures? Can you provide a number?*

One question of our study is to estimate how much water vapour has to be enhanced to allow chlorine activation at higher temperatures. For our standard case 10.6 ppmv $H_2O$ are necessary for the occurrence of chlorine activation. To clarify that in the text, we added the following sentence to the introduction.

An aim of our study is to investigate how much water vapour has to be enhanced for chlorine activation to occur at these higher temperatures.

*- Model set-up: Why do you neglect NAT and ice in this study? NAT is probably not an issue, but how about ice? Especially since you mention a potential impact of ice particles on the water vapour threshold for Cl activation on liquid aerosol particles due to different heterogeneous reactivities (p32, l3-5).*

We decided to calculate heterogeneous chemistry only on liquid particles to have a better comparability with the studies of Anderson et al. (2012) and Anderson and Clapp (2018). In their studies, they investigate the impact of binary $H_2O$/sulphate aerosols on mid-latitude ozone chemistry. To clarify this in the text, we added the following sentence to Sec. 2 (Model Setup).

In contrast to the setup in Grooß et al. (2011), Müller et al. (2018) and Zafar et al. (2018), only formation of liquid particles (both binary $H_2O/H_2SO_4$ and ternary $HNO_3/H_2O/H_2SO_4$ solutions) is allowed (i.e. no NAT or ice particles are formed in this model setup) to enable a better comparability with the studies of Anderson et al. (2012, 2017) and Anderson and Clapp (2018).

In our study, we mention that chlorine activation on ice has been observed and analysed in former studies (p.31 l. 29-31, p.33, l.12–15, ACPD version). We decided to not insert a further case study, which additionally shows the

impact of ice formation, to provide a better readability. We think that to reasonably show the impact of ice formation on the ozone process, however a comprehensive study would be necessary, which is beyond the slope of our current study.

*- Model set-up: Could provide a short description of the treatment of liquid aerosol particles in your model? In Table 1 you provide the gas phase equivalent H2SO4 mixing ratio and Fig. 4 ff show the surface area density, but some more information would be great. For example, which H2SO4 wt% in the liquid aerosol particles is reached under the assumed high water vapour/low temperature conditions?*

As proposed, we included a description of liquid aerosol treatment in Sec. 2 (P.4, l. 8, revised version of the manuscript).

For heterogeneous particle formation, the initial liquid aerosol number density ($N_0$=10.0 cm$^{-3}$), the standard deviation of the logarithmic normal distribution of the particle size ($\sigma$=1.8), and the gas phase equivalent of the amount of sulfuric acid in the aerosol (for chosen values see Tab. 1) are set prior to the simulation. The gas phase equivalent is used to calculate the density of liquid particles as described in the study of Shi et al. (2001) (binary solutions) and Luo et al. (1996) (ternary solutions). Particle size and surface area densitiy are calculated based on the density of liquid particles, the areosol number density, and the standard deviation.

Furthermore, we added an additional plot (in Fig. 8, revised version of the manuscript) as shown in Fig. 1 of this reply, which presents the $H_2SO_4$ wt% depending on the water vapour mixing ratio. The following description is included in Section 4.2 (p.19, l.19 revised version of the manuscript).

The particles $H_2SO_4$ wt% decreases for all cases with increasing water vapour from more than 50 wt% at 5 ppmv $H_2O$ to around 20 wt% at 20 ppmv $H_2O$ due to an inceasing uptake of $H_2O$ in the thermodynamic equilibrium.

[Figure]

Figure 1: Dependence on water vapour of the rate of the the main heterogeneous chlorine activation reaction R1, the rate coefficient ($k_{R1}$), the $\gamma$-value $\gamma_{R1}$, the liquid surface area density $A_{liq}$, and the $H_2SO_4$ weight %. Presented parameters correspond to the values after the first chemistry time step of the box-model simulation. Additionally the impact of an enhanced sulphate content ($0.8\,\mathrm{ppbv}\ H_2SO_4$, yellow), reduced $NO_y$ ($0.8\ NO_y$, green), reduced $Cl_y$ ($0.8\ Cl_y$, black) and enhanced temperatures (red) is shown. The standard case is shown as blue squares.

*- P5, l32/33: What was the overall background H2O mixing ratio for the calculated trajectories? And temperature? By how much are the selected trajectories colder?*

In the Figure 2 of this reply, we illustrate the water vapour and temperature conditions of the SEAC$^4$RS measurements and the calculated backwards trajectories. The top panel shows the distibution of measured water vapour and temperature during the SEAC$^4$RS campaign at a pressure of less than 100 hPa of all flights. This pressure range was chosen to select stratospheric measurements. The overall background water vapour is around 6–7 ppmv.

We calculated trajectories starting at the location of the measurements with enhanced water vapour of at least 10 ppmv $H_2O$. The temperature along these backward trajectories (in total 294) is shown in the bottom panels. The left panel shows the mean temperature and the right panel the minimal temperature of each calculated trajectory. The temperature of our standard trajectory is marked with a red arrow. Most of the trajectories have a mean temperature of 201–202 K and a minimal temperature of 199–200 K. Our standard trajectory has a mean temperature of 199 K and a minimal temperture of 197 K. In total, 28 trajectories with similar low temperatures were calculated. Most of these trajectories correspond to higher measured $CH_4$ mixing ratios and thus very low $Cl_y$ mixing ratios. Only three of the trajectories with very low temperatures and enhanced water vapour correspond to measurements with chlorine high enough to show a visible impact on ozone chemistry.

We added in Sec. 2.2 (p.5, l.1–4, revised version of the manuscript) as follow that $Cl_y$ was a criteria for choosing our standard trajectory in addition to the water vapour mixing ratio and temperature.

A selected example of calculated trajectories is shown in Fig. 1 (revised version). This trajectory was chosen for the chemical analysis, because its initial conditions exhibited enhanced water vapour relative to the overall background, low temperatures and enhanced $Cl_y$ (higher than for comparable water vapour and temperature conditions). $Cl_y$ was calculated from tracer-tracer correlations (see Sec. 2.3).

[Figure]

Figure 2: Top panel: Frequency distribution of water vapour mixing ratios and temperature measured during all flights of the SEAC[4]RS campaign at a pressure level of less than $100\,\mathrm{hPa}$. The colour scheme shows the number of measurements in that temperature- and water vapour regime. Bottom panels: Temperature along all backward trajectories (calculated) started at the location of a measurement with a water vapour mixing ratio lager than $10\,\mathrm{ppmv}$. On the y-axis, the total number of trajectories in that temperature regime is plotted. The mean temperature (left) and the minimum temperature (right) of each trajectory is shown.

*- Sect. 2.3 Initialization: In Table 1 it is mentioned that the ClOx species were initialized with 0. Ok, in the mid-latitude lower stratosphere most chlorine is deactivated, but I am wondering how your initial phase in Fig. 4 would look like, if the trajectories had started with non-zero ClOx? Maybe you could add a short discussion.*

We conducted a further simulation, assuming $ClO_x$ counts $1\%$ of total $Cl_y$. This yields slightly more $ClO_x$ in the initial phase. But it changes neither the rate of R1 nor the water vapour threshold or the time, when ozone destruction starts significantly. We added a small discussion in Sec. 2.3.1 after

the sequence about $Cl_y$ initialization (P.6, l.13–15, revised version of the manuscript).

Since most $Cl_y$ is deactivated in the mid-latitude lowermost stratosphere, the initial mixing ratio of $ClO_x$ species is assumed to be zero. A simulation assuming a ClO mixing ratio of 1% of total $Cl_y$, does not yield a significant difference to our standard case.

*- Sect. 2.3.3: I do not really see the point of discussing the MACPEX case. In my view it is just another special case that could be covered by the sensitivity simulations. For the sake of clarity, I would suggest to leave it away.*

We agree with the reviewer, that the discussion of the MACPEX case is neither enhancing the readability of the paper nor leading to other results than our standard case based on SEAC[4]RS measurements. However, we think that presenting a further case, which yields similar sensitivities as the SEAC[4]RS simulations, complements our results. Hence, we decided to present the MACPEX case in the appendix and adapted the figures and the text in Sec. 2. In Section 4.1, the last sequence (P. 18, l. 4–8, revised version of the manuscript) was changed, but still refers to the MACPEX case:

As a further example for an event with high stratospheric water vapour mixing ratios based on airborne measurements, simulations based on measurements during the Mid-latitude Airborne Cirrus Properties Experiments (MACPEX) were conducted. This campaign was based in Texas during springtime 2011 and hence prior to the formation of the North American Monsoon (NAM). A detailed description of this MACPEX case is given in the appendix (A1). For the MACPEX case, changes in sulphate, $Cl_y$ and $NO_y$ mixing ratios affect the water vapour threshold similarly to that observed for the SEAC[4]RS trajectory. Thus, the MACPEX results confirm the SEAC[4]RS findings. Therefore, we conclude that in the considered temperature range ($\sim$197–202 K), an ozone reduction occurs after exceeding a water vapour threshold and that this threshold varies with $Cl_y$, $NO_y$, sulphate content and temperature.

*- Fig. 2: What is the rationale behind the chosen H2O mixing ratios of 5 and 15 ppmv? This choice seems a bit arbitrary to me. Why do you not use the 10.6 ppmv H2O of your standard case?*

This study is aimed to investigate the sensitivity of mid-latitude lowermost stratospheric ozone chemistry to water vapour and to analyse the chemical mechanisms in detail. The best way to show the differences between the ozone chemistry at low and high water vapour mixing ratios is to choose water vapour mixing ratios, which are clearly in that water vapour regime. To be able to show the chlorine activation phase, we decided to take a water vapour mixing ratio of 15 ppmv instead of 20 ppmv. At 20 ppmv chlorine activation is very fast and reaction rates and mixing ratios are changing strongly. Since the mixing ratio of 10.6 ppmv $H_2O$ of the realistic case is elevated comparing to the overall background conditions and close to the threshold, we decided against taking this case to illustrate the chemistry in a low water vapour regime.
To clearify this decision we added the following sentence in Sec. 3 (P.8, L.12, revised version of the manuscript).

These water vapour mixing ratios are chosen, because they are clearly in the regime of the low water vapour background (5 ppmv) of the lower mid-latitude stratosphere and of enhanced water vapour (15 ppmv) as it can be reached through convective overshooting events.

Furthermore, we clarified that choice in the description of Fig.2.

These water vapour mixing ratios are chosen, because they are clearly in the regime of low (5 ppmv) and elevated (15 ppmv) water vapour.

*- P10, discussion of Fig. 3: In line 3-5 the water vapour threshold is defined as H2O mixing ratio "at which the end ozone value clearly falls below the end ozone that is reached for low water vapour amounts." What is meant by "clearly" ? That sounds a bit vague. Can you be a bit more quantitative? Furthermore, it is stated that "by 12 ppmv of water vapour, the system is clearly in an ozone destruction regime." But is 12 ppmv not just the water vapour mixing ratio at which the transition between ozone production to ozone destruction occurs? At lower H2O values, end ozone is higher than initial ozone.*

We agree with the reviewer that the description of the threshold is vague formulated. To make the formulation of the threshold more clear, we determined the lowest water vapour mixing ratio at which chlorine activation occurs to be the threshold. We changed the discussion of Fig. 3 as follows:

Blue squares lying above that line are cases with ozone production, those lying below that line are cases with ozone destruction. The decrease of final ozone with higher water vapour mixing ratios is related to chorine activation. The time until chlorine activation occurs in the simulation is plotted in Fig. 3 as violet triangles, assuming that chlorine activation occurs when the $ClO_x$ mixing ratio exceeds 10% of total $Cl_y$ (Drdla and Müller, 2012). Shown is the time when chlorine activation first occurs in the model. Since the $ClO_x/Cl_y$ ratio is dependent on the diurnal cycle, the 24-hours mean value of the $ClO_x$ mixing ratio was used to determine the chlorine activation time. For low water vapour mixing ratios, no chlorine activation time is plotted, because no chlorine activation occurs. For chlorine activation to occur, a threshold in water vapour has to be reached. Here, we determine the lowest water vapour mixing ratio at which chlorine activation occurs as water vapour threshold (marked by a blue arrow in Fig. 3). In our standard case, this threshold is reached at a water vapour mixing ratio of 10.6 ppmv. Between 10.6 and 12 ppmv $H_2O$, chlorine activation leads not to an ozone destruction during the 7-day simulation. For 10.6 to 11.2 ppmv $H_2O$, chlorine only remains activated for up to 28 h, because of increasing temperatures, and almost no impact on final ozone is observable. By 12 ppmv of water vapour, chlorine activation yields ozone destruction within the 7-day simulation. Near the water vapour threshold, the activation time is 24 to 36 hours and it decreases with increasing water vapour mixing ratios. It requires only 5 hours at 20 ppmv $H_2O$.

*-Fig. 4b: Where does the first sharp peak in the light blue line (ClONO2+HCl) come from?*

In the chosen standard case (15 ppmv), the chlorine activation reaction R1 already occurs significantly within the first diurnal cycle. This would yield an increase in $ClO_x$. Since the $NO_x$ mixing ratio is high at the same time, every formed $ClO_x$ leads to the formation of $ClONO_2$. Hence, the $ClONO_2$ mixing ratio increases (Fig. 4f) and thus the rate of R1 ($ClONO_2$+HCl). The chlorine activation chain (P.13, l. 3–7, ACPD version of the manuscript) requires light to transform $Cl_2$ to Cl-radicals. During night $ClO_x$ is accumulated in $Cl_2$ and $Cl_2O_2$. Hence, there is no ClO from which $ClONO_2$ could be formed

and the rate of R1 decreases rapidly. This yields the first sharp peak in the light blue line in Fig. 4b. For a better readability, we decided to not add this discussion to the paper.

*- P13, l20ff: Shouldn't this sentence read: Dependent on temperature and water vapour content, the HNO3 formed remains in the condensed phase.? And further down: … 64% of the HNO3 remains in the condensed phase on the day with the lowest temperature, while at higher temperatures… 85% of HNO3 are released to the gas phase.… Is the HNO3 shown in Fig. 4d gas-phase only or total HNO3?*

As the reviewer recommended, we revised the text as follows:

Dependent on temperature and water vapour content, the $HNO_3$ formed remains in the condensed particles. In the standard simulation using $15\,\mathrm{ppmv}$ $H_2O$, 64% of $HNO_3$ remains in the condensed phase on the day with the lowest temperature ($197.3\,\mathrm{K}$, 2 Aug 2013), while at higher temperatures (4–7 August 2013) 85% of $HNO_3$ are released to the gas phase.

To clarify that $HNO_3$ in Fig. 4d is total $HNO_3$ (gas phase + condensed), we revised the description of Fig. 4.

Panels (d), mixing ratio of $HNO_3$ (gas phase + condensed), $NO_x$ and $ClONO_2$, ...

*- P18/19 cycles C7 and C8: I have a hard time to follow the construction of these reaction cycles. My impression is that the authors combined different reactions until they ended up with the intended net reaction. For example: In R29 NO2 is formed, which in the next step photolyzes (R15). Lower down another NO2 is formed (R32), but this time it reacts with ClO to ClONO2 (R22).*

We revised the part about the maintenance of activated chlorine in Sec. 3.2.2. The detailed description of the pathways C7 and C8 is moved to the appendix. In Sec. 3.2.2, it is substituted by a scheme, which focuses on the main reactions to balance chlorine activation and deactivation as well as $HNO_3$ and $NO_x$. This scheme is shown in Fig. 3 of this reply.

[Figure]

Figure 3: Reaction scheme to illustrate the balance between chlorine activation and chlorine deactivation (blue, right) and $NO_x$ activation and deactivation (green, left). The heterogeneous reaction $ClONO_2+HCl$ (R1) links both cycles. Additional reaction pathways, which balance radicals are shown in light colours.

*- P20, l 14-16: I think it would be helpful to mention the reason for the lower H2O threshold under enhanced sulphate conditions, namely the increased SAD. In general, the presentation of the model results would benefit from some more explanation of the underlying processes. This holds also for Sect. 4.2 and the sensitivity of the uptake coefficient to H2O or temperature. By the way, why is the 10xH2SO4 case not shown in Fig. 7?*

The sensitivity of the water vapour threshold to temperature, sulphate, $Cl_y$ and $NO_y$ is described in Sec. 4.1 and explained in Sec. 4.2. To clarify this, we added a reference to Section 4.2 in Sec. 4.1 (P.18, l.2, revised version of the manuscript).

The sensitivity of the water vapour threshold to temperature, sulphate abundance and $Cl_y$ and $NO_y$ mixing ratio is explained in Section 4.2.

*... In general, the presentation of the model results would benefit from some more explanation of the underlying processes. This holds also for Sect. 4.2*

*and the sensitivity of the uptake coefficient to H2O or temperature. ...*

To go into the underlying processes, we mention that the $\gamma$-value depends on the weight-% on sulphuric acid in the particles (which is known from laboratory studies). Therefore a further panel illustrating the dependence of the $H_2SO_4$ wt% on the water vapour mixing ratio was included in Fig. 8 of the revised version of the manuscript (Fig. 1 of this reply). We revised the text in Sec. 4.2 (p.19, l.11–23, revised version of the manuscript):

The $\gamma$-value describes the uptake of $ClONO_2$ into liquid particles due to the decomposition of $ClONO_2$ during reaction R1 and is thus a measure of the probability of the occurrence of this heterogeneous reaction (Shi et al., 2001). Laboratory studies showed a dependence of $\gamma_{R1}$ on the solubility of HCl in the droplet, which generally increases for a lower $H_2SO_4$ fraction in the particle ($H_2SO_4$ wt%) (Elrod et al., 1995; Hanson, 1998; Zhang et al., 1994; Hanson and Ravishankara, 1994). From Eq. 2 it is obvious that a large surface area $A_{liq}$ and a high $\gamma$-value $\gamma_{R1}$ increase $k_{R1}$ and thus the heterogeneous reaction rate $v_{R1}$.
In Figure 8, the impact of the water vapour content on the $H_2SO_4$ weight-percent, $\gamma_{R1}$, $A_{liq}$, $k_{R1}$ and the reaction rate $v_{R1}$ is shown. To avoid the influence of R1 itself on these parameters as much as possible, these parameters are selected for 1 August 2013 at 13:00 UTC. This point in time corresponds to the values after the first chemistry time step during the chemical simulation. The particles $H_2SO_4$ wt% decreases for all cases with increasing water vapour from more than 50 wt% at 5 ppmv $H_2O$ to around 20 wt% at 20 ppmv $H_2O$ due to an increasing uptake of $H_2O$ in the thermodynamic equilibrium. The standard case is illustrated in blue squares (Fig. 8) and exhibits a strongly increasing gamma value especially for water vapour mixing ratios between 9 and 14 ppmv due to a lower $H_2SO_4$ wt%. In the same water vapour range, the liquid surface area density $A_{liq}$ increases slightly. It increases more for higher water vapour mixing ratios because of $HNO_3$ uptake into the particles.

*...By the way, why is the 10xH₂SO₄ case not shown in Fig. 7?*

The aim of Fig. 7 (ACPD version of the manuscript, Fig. 8 of revised version of the manuscript, Fig 1 in this reply) is to illustrate the sensitivity of the water vapour threshold. To focus on this processes, we decided to not

show the $10xH_2SO_4$ case here. The $10xH_2SO_4$ case yields higher values for the surface area, the reaction rate and the rate constant. If we would show also the $10xH_2SO_4$ case, the detailed discussion in Sec. 4.2 would not benefit from Fig. 7 (ACPD version of the manuscript, Fig. 8 of revised version of the manuscript).

*- P20, discussion of Fig. 6b: I do not understand the statement that similar conclusions as for the SEAC$^4$RS trajectory hold for the MACPEX trajectory. For almost all cases final $O_3$ is higher than initial $O_3$. And why is there only a subset of sensitivities shown for the MACPEX trajectory? Why not also +1K and $10xH_2SO_4$?*

The statement refers to the sensitivity of the water vapour threshold (and thus chlorine activation) to $Cl_y$, $NO_y$, temperature and sulphate. This sensitivity is similar in the MACPEX and SEAC$^4$RS case. In the MACPEX case, final ozone is higher than initial ozone because of the low $Cl_y$ mixing ratio ($\sim$55 pptv). Hence, the catalytic ozone loss cycles have lower rates and ozone is destroyed slowlier. We added the $10xH_2SO_4$ and $+1\,K$ case to Fig. 6b (ACPD version of the manuscript, Fig. B2 in the revised version of the manuscript).

*- Fig. 6: What is the rationale for the $3xH_2SO_4$ case?*

The $3\times$ $H_2SO_4$ case is shown here as well, because it is the lowest $H_2SO_4$ amount, which would yield ozone destruction in the realistic case (see Sec. 5.2).

*- P25, l3-5: I do not fully understand this argumentation. Do you mean that there is no longer enough ClONO2 available to react with the HCl taken up by the condensed particles and that therefore the enhanced HCl uptake does not lead to further chlorine activation and ozone loss?*

This argumentation only refers to ozone destruction, not also chlorine activation. After chlorine activation occurred, ozone loss is driven by chlorine catalysed ozone loss cycles. The more chlorine is available in the gas phase, the more ozone can be destroyed by this cycles. An uptake of HCl into the condensed particles reduces gas phase chlorine. Hence, also the chlorine catalysed ozone loss cycles have lower rates and less ozone is destroyed.

*- Fig. 8: This figure nicely shows the different ozone regimes as a function of temperature and H2O for two different sulphate conditions. Would it possible to provide such figure also for Cly, Bry, NOy sensitivities? I am aware that this is a multi-dimensional problem, but I think it would be really helpful to get an overview under which conditions ozone formation and ozone depletion occurs. This would also help to get a better understand of the importance of this mechanism on larger scales.*

We did the same plot for the $0.8\,\mathrm{Cl}_y$, $0.8\,\mathrm{NO}_y$ and $0.5\mathrm{Br}_y$ case (see Fig. 5 of this reply). Only minor differences regarding the standard case are observable. For the $0.8\,\mathrm{Cl}_y$ and the $0.5\,\mathrm{Br}_y$ case, less ozone is destroyed if chlorine activation occurs and the chlorine activation line is slightly shifted. Assuming the $0.8\,\mathrm{NO}_y$ case yields a minor shift of the chlorine activation line. Since these tendencies are rather small, we think they are more clear in Fig. 7 of the revised version of the manuscript. Hence, we decided against showing these plots in the paper.

We agree with the reviewer that it would be helpful to apply the processes analysed here on a larger scale also showing the climatological perspective. Therefore, we started a second comprehensive study looking on the likelihood of the occurrence of ozone loss in the WACCM model at mid-latitudes in the lowermost stratosphere that is beyond the slope of our current study. The study here is mainly focused on the mechanisms itself. To enable here already an estimation for the relevance of this process under further conditions, we conducted the sensitivity studies ($0.8\mathrm{Cl}_y$, $0.8\mathrm{NO}_y$, $0.5\,\mathrm{Br}_y$, several $\mathrm{H_2SO_4}$ abundances and temperatures) and case studies (other time duration, observed SEAC[4]RS conditions, MACPAX case). These additional studies give an estimation for extending the results of this study to a lager scale.

[Figure]

Figure 4: Relative ozone change during the 7-day simulation along the standard trajectory dependent on temperature and $H_2O$ mixing ratio for several conditions. The white line corresponds to the water and temperature dependent chlorine activation threshold. In the top-panels, climatological non enhanced (left panel) and enhanced (right panel) sulphate conditions are shown. In the middle panels the impact of a reduction of $Cl_y$ (left) and $NO_y$ (right) and in the bottom panel of $Br_y$ on the ozone change are shown.

*- Sect. 5: As mentioned above I think the case studies are too much and do not help to provide further insights into the main chemical mechanism. Its rather confusing to distinguish the various atmospheric compositions assumed for the individual model simulations. In particular the 19-days case seems*

*trivial to me: longer time under cold and humid conditions, more ozone loss.*

We agree that the sensitivity studies are very detailed and comprehensive. But we think, that they give an insight in the sensitivities of the ozone loss processes and thus enable to apply this processes on a lager scale of realistic conditions than only the chosen example. To make this part less excessive, we moved the chemical details of the 'Case of high $Cl_y$', the 'Reduced $Br_y$ case' and the '19-day Simulation' to supplemental material. We only show the effect of this cases on the water vapour threshold in a new Figure (see Fig. 5 of this reply, Fig. 11 of the revised version of the manuscript). We did not change the 'Case based on observations'.

*Technical corrections:*
*- P3, l19: hypothesis hat $\rightarrow$ hypothesis that*
*- P4, l8: ans $\rightarrow$ and*
*- P4, l20: HNO3+2N2O5*
*- P8, l3: N2O, 2 should be subscript*
*- P9, l28: ClOx, x should be subscript*
*- P16, l3: simultanous $\rightarrow$ simultaneous*
*- P17, l29: HCl-formating $\rightarrow$ HCl-forming*
*- P34, l19: and extreme $\rightarrow$ an extreme*

As the reviewer recommended, we revised these sequences in the text.

*- Units: The text is a mixture of ppb/ppt and ppbv/pptv. I assume it is always volume mixing ratio?*

We revised the text to use only the volume mixing ratio.
*- Chemical reactions: It would be nice to mark heterogenous reactions as such.*

In the revised version of the manuscript, heterogeneous reactions are marked ($\xrightarrow{het.}$).

*- Table 1: I would suggest to add Cly and NOy for completeness. And are 0.6 ppbv and 2.0 ppbv H2SO4 not also sensitivity simulations?*

[Figure]

Figure 5: The water dependent final ozone value is shown for (a) the "Case of high $Cl_y$" (see Tab. 1 for $NO_y$ and $Cl_y$ initialisation) assuming background aerosol (light blue) and tripled $H_2SO_4$ (yellow), (b) reduced $Br_y$ (light blue, "Reduced $Br_y$ case"), and (c) an extended time period of activated chlorine (light blue, "19-day simulation"). In panel (b) and (c) also final ozone of the standard case is shown (blue). Initial ozone is marked with a grey line. Note that the scale of all y-axis differ.

As proposed by the reviewer, we added $Cl_y$ and $NO_y$ to Table 1 and moved the initialisations with enhanced $H_2SO_4$ (0.6 and 2.0 $H_2SO_4$) to the column, which presents the sensitivity studies.

*- Fig 1.: It would be nice to mark the time of measurement by a red square or a vertical line also to the left panels. And maybe the tropopause altitude at the location of the airparcel could also be added to the left panel.*

As recommended by the reviewer, we mark the time of measurement in the left panels of Fig. 1 with a vertical red line as well as the tropopause at the location of the air parcel with a horizontal grey line.

*- Fig. 4, 9, 10, 11, 12: Which time of day do the x-axis tick marks refer to? One can infer night-/day-time from the reactions with OH, but it would be helpful to provide this information, e.g., in the caption.*

In the ACPD version of the manuscript, the x-axis tick marks refer to 00:00 UTC. In the revised version of the manuscript, they refer to 00:00 local time, which corresponds to 06:00 UTC on that day. To clarify this in the text, we added the following sentence to the caption of that figures.

The x-axis ticks refer to 00:00 local time (06:00 UTC).

*- Fig. 7: units (y-axis) are missing*

We added the units in Fig. 7 in the ACPD version of the manuscript (Fig. 8 of the revised version of the manuscript, Fig. 1 in this reply). The $\gamma$-value has no unit.

**References**

Anderson, J. G. and Clapp, C. E.: Coupling free radical catalysis, climate change, and human health, Phys. Chem. Chem. Phys., 20, 10 569–10 587, doi:10.1039/C7CP08331A, 2018.

Anderson, J. G., Wilmouth, D. M., Smith, J. B., and Sayres, D. S.: UV Dosage Levels in Summer: Increased Risk of Ozone Loss from Convectively Injected Water Vapor, Science, 337, 835–839, doi:10.1126/science.1222978, 2012.

Anderson, J. G., Weisenstein, D. K., Bowman, K. P., Homeyer, C. R., Smith, J. B., Wilmouth, D. M., Sayres, D. S., Klobas, J. E., Leroy, S. S., Dykema, J. A., and Wofsy, S. C.: Stratospheric ozone over the United States in summer linked to observations of convection and temperature via chlorine

and bromine catalysis, Proc. Natl. Acad. Sci., 114, E4905–E4913, doi: 10.1073/pnas.1619318114, 2017.

Drdla, K. and Müller, R.: Temperature thresholds for chlorine activation and ozone loss in the polar stratosphere, Ann. Geophys., 30, 1055–1073, doi:10.5194/angeo-30-1055-2012, 2012.

Elrod, M. J., Koch, R. E., Kim, J. E., and Molina, M.: HCl vapour pressures and reaction probabilities for $ClONO_2$+HCl on liquid $H_2SO_4$-$HNO_3$-HCl-$H_2O$ solutions, Faraday Discuss., 100, 269–278, 1995.

Grooß, J.-U., Brautzsch, K., Pommrich, R., Solomon, S., and Müller, R.: Stratospheric ozone chemistry in the Antarctic: What controls the lowest values that can be reached and their recovery?, Atmos. Chem. Phys., 11, 12 217–12 226, doi:10.5194/acp-11-12217-2011, 2011.

Hanson, D. R.: Reaction of $ClONO_2$ with $H_2O$ and HCl in sulfuric acid and $HNO_3$/$H_2SO_4$/$H_2O$ mixtures, J. Phys. Chem. A, 102, 4794–4807, 1998.

Hanson, D. R. and Ravishankara, A. R.: Reactive Uptake of $ClONO_2$ onto Sulfuric Acid Due to Reaction with HCl and $H_2O$, J. Phys. Chem., 98, 5728–5735, 1994.

Luo, B., Krieger, U. K., and Peter, T.: Densities and refractive indices of $H_2SO_4$/$HNO_3$/$H_2O$ solutions to stratospheric temperatures, Geophys. Res. Lett., 23, 3707–3710, doi:10.1029/96GL03581, 1996.

Müller, R., Grooß, J.-U., Zafar, A., Robrecht, S., and Lehmann, R.: The maintenance of elevated active chlorine levels in the Antarctic lower stratosphere through HCl null cycles, Atmos. Chem. Phys., 18, 2985–2997, doi: 10.5194/acp-18-2985-2018, 2018.

Shi, Q., Jayne, J. T., Kolb, C. E., Worsnop, D. R., and Davidovits, P.: Kinetic model for reaction of $ClONO_2$ with $H_2O$ and HCl and HOCl with HCl in sulfuric acid solutions, J. Geophys. Res., 106, 24 259–24 274, doi: 10.1029/2000JD000181, 2001.

Solomon, S.: Stratospheric ozone depletion: A review of concepts and history, Rev. Geophys., 37, 275–316, doi:10.1029/1999RG900008, 1999.

Vogel, B., Feck, T., and Grooß, J.-U.: Impact of stratospheric water vapor enhancements caused by $CH_4$ and $H_2$ increase on polar ozone loss, J. Geophys. Res., 116, D05301, doi:10.1029/2010JD014234, 2011.

Zafar, A. M., Müller, R., Grooß, J.-U., Robrecht, S., Vogel, B., and Lehmann, R.: The relevance of reactions of the methyl peroxy radical (CH3O2) and methylhypochlorite (CH3OCl) for Antarctic chlorine activation and ozone loss, Tellus B, 70, 1507 391, doi:10.1080/16000889.2018.1507391, 2018.

Zhang, R., Jayne, J. T., and Molina, M. J.: Heterogeneous interactions of $ClONO_2$ and HCl with sulfuric acid tetrahydrate: Implications for the stratosphere, J Phys Chem, 98, 867–874, 1994.